# Learning Cocoercive Conservative Denoisers via Helmholtz Decomposition for Poisson Imaging Inverse Problems

**Deliang Wei**
School of Mathematical Sciences,
East China Normal University,
Shanghai 200241, China
52215500006@stu.ecnu.edu.cn

**Peng Chen**
School of Mathematical Sciences,
East China Normal University,
Shanghai 200241, China
52265500005@stu.ecnu.edu.cn

**Haobo Xu**
School of Mathematical Sciences,
East China Normal University,
Shanghai 200241, China
52275500009@stu.ecnu.edu.cn

**Jiale Yao**
School of Mathematical Sciences,
East China Normal University,
Shanghai 200241, China
52285500019@stu.ecnu.edu.cn

**Fang Li**[*]
School of Mathematical Sciences,
Key Laboratory of MEA
(Ministry of Education),
Shanghai Key Laboratory of PMMP,
East China Normal University,
Shanghai 200241, China
fli@math.ecnu.edu.cn

**Tieyong Zeng**
Institute for Advanced Study,
Beijing Normal-Hong Kong Baptist University,
Zhuhai 519000, Guangdong, China
School of Mathematics and Statistics,
Guangzhou Nanfang College,
Guangzhou 510970, Guangdong Province, China
tieyongzeng@uic.edu.cn

## Abstract

Plug-and-play (PnP) methods with deep denoisers have shown impressive results in imaging problems. They typically require strong convexity or smoothness of the fidelity term and a (residual) non-expansive denoiser for convergence. These assumptions, however, are violated in Poisson inverse problems, and non-expansiveness can hinder denoising performance. To address these challenges, we propose a cocoercive conservative (CoCo) denoiser, which may be (residual) expansive, leading to improved denoising performance. By leveraging the generalized Helmholtz decomposition, we introduce a novel training strategy that combines Hamiltonian regularization to promote conservativeness and spectral regularization to encourage cocoerciveness. We prove that CoCo denoiser is a proximal operator of a weakly convex function, enabling a restoration model with an implicit weakly convex prior. The global convergence of PnP methods to a stationary point of this restoration model is established. Extensive experimental results demonstrate that our approach outperforms closely related methods in both visual quality and quantitative metrics. A test code is provided for reproducibility[2].

---

[*]Corresponding author
[2]https://github.com/FizzzFizzz/CoCo-PnP

# 1  Introduction

Image restoration is a fundamental task in the computer vision field. Although most works assume that the noise follows Gaussian distribution [43, 12], under low-light condition, such as hyperspectral imaging [53, 17], medical imaging [67], and limited photon imaging [13], images are inevitably corrupted by Poisson noises. In this paper, we focus on solving Poisson inverse problems by Plug-and-Play (PnP) methods.

The following formula describes the Poisson corruption procedure:

$$f \sim \text{Poisson}(pKu)/p, \tag{1}$$

where $u$ is the potential image, $f$ is the noisy image, the peak value $p$ is the average number of photons recieved per pixel, and $K$ is a known linear operator such as identity I, blur kernel, or Radon transform. A small $p$ corresponds to a severe noise environment. In order to recover $u$ from $f$, a variational approach is considered:

$$\arg\min_{u \in V} F(u) + G(u), G(u) = \lambda \langle \mathbf{1}, Ku - f \log Ku \rangle, \tag{2}$$

where $V$ is the underlying Hilbert space endowed with the inner product $\langle \cdot, \cdot \rangle$, $F$ denotes the prior regularization term, and the data fidelity $G$ is the Bregman divergence, with $\lambda > 0$ being the balancing parameter. Typical choices for $F$ includes total variation [59] and its variants [33, 11], weighted nuclear norm [23], and group-based low rank prior [44]. First order methods are employed to find $\hat{u}$, such as the proximal gradient descent method (PGD):

$$u^{k+1} = \text{Prox}_{\frac{F}{\beta}}(u^k - \beta^{-1} \nabla G(u^k)), \tag{3}$$

and the alternating direction method with multipliers (ADMM) [10]:

$$\begin{aligned}
u^{k+1} &= \text{Prox}_{\frac{F}{\beta}}(v^k - b^k), \\
v^{k+1} &= \text{Prox}_{\frac{G}{\beta}}(u^{k+1} + b^k), \\
b^{k+1} &= b^k + u^{k+1} - v^{k+1},
\end{aligned} \tag{4}$$

where $\beta > 0$ is a balancing parameter. For a closed, convex and proper (CCP) function $F : V \to \bar{\mathbb{R}} = (-\infty, \infty]$, the proximal operator $\text{Prox}_F : V \to V$ is defined as: $\text{Prox}_F(y) := \arg\min_{x \in V} F(x) + \frac{1}{2}\|x - y\|^2$.

**Plug-and-play methods.** By replacing $\text{Prox}_{\frac{F}{\beta}}(\cdot)$ with an off-the-shelf Gaussian denoiser $\text{D}_\sigma(\cdot)$, where $\sigma$ denotes the denoising strength, Venkatakrishnan et al. introduced the plug-and-play ADMM (PnP-ADMM) algorithm [69]. Since then, PnP methods have achieved surprisingly remarkable recovery results in many inverse problems, including bright field electron tomography [64], diffraction tomography [65], camera image processing [25], low-dose CT imaging [9, 72], Poisson image denoising [38, 27], deblurring [36], inpainting [77], and super-resolution [37, 32].

**Challenges.** Despite the impressive recovery performance, the convergence analysis of PnP methods remains challenging, especially in the context of Poisson inverse problems. The standard convergence of ADMM and PGD relies on the convexity of both $F$ and $G$. However, a convex prior term $F$ often leads to limited recovery performance. When $F$ is weakly convex, additional assumptions on $G$, such as strong convexity or smoothness, are required. Unfortunately, these assumptions do not hold in Poisson inverse problems. Moreover, in general, a deep denoiser is not a proximal operator, which further complicates the situation.

In order to get a proximal $\text{D}_\sigma$, it should satisfy some Lipschitz properties such as non-expansiveness, and be the (sub)gradient of some CCP potential $\phi$ [47, 22]. In the following, we briefly discuss approaches to satisfy these conditions, as well as some related convergent PnP methods.

**Lipschitz property.** There are mainly two methods to ensure the Lipschitz property: **real spectral normalization (RealSN)** and **spectral regularization (SR)**. Ryu et al. enforced a contractive $\text{I} - \text{D}_\sigma$ by applying RealSN to the denoiser, which normalized the spectral norm of each layer [60]. However, RealSN is computationally expensive, and was specifically designed for denoisers with cascade residual learning structures, such as DnCNN [74], making it unsuitable for other networks like UNet [58]. A more flexible approach is SR technique introduced by Terris et al. [66]. In order to get a firmly non-expansive $\text{D}_\sigma$, they added a spectral term $\min\{1 - \epsilon, \|2\,\text{J} - \text{I}\|_*\}$ to the original loss

function, such that $\|2\,\mathrm{J}-\mathrm{I}\,\|_* \leq 1$, where $\mathrm{J} = \nabla\,\mathrm{D}_\sigma$ is the Jacobian matrix of $\mathrm{D}_\sigma$, and $\epsilon \in (0,1)$ is a pre-defined parameter. With this spectral constraint, $\mathrm{D}_\sigma$ is firmly non-expansive, and thus becomes the resolvent of some implicitly maximally monotone operator (RMMO), incorporated into the Douglas-Rachford splitting (DRS) method. RMMO-DRS only requires a convex fidelity, making it suitable for Poisson inverse problems. Inspired by SR, Wei et al. proposed to train a strictly pseudo-contractive (SPC) denoiser, termed SPC-DRUNet [71]. By incorporating the Ishikawa process and half-quadratic splitting (HQS), they developed the PnPI-HQS method. The convergence of PnPI-HQS only requires a convex fidelity term $G$, which makes it suitable for Poisson problems. SPC is much weaker than firm or residual non-expansiveness. However, SPC-DRUNet is in general not a proximal operator, and PnPI-HQS does not solve any optimization problem.

**Conservativeness.** To ensure the conservative property, many works attempt to explicitly define a prior function $\phi$, such that the denoiser is its gradient or proximal operator. Romano et al. proposed the regularization by denoising (RED) [57]. The RED prior term takes the form of $\phi(x) = \frac{1}{2}\langle x, x - \mathrm{D}_\sigma(x)\rangle$. Under the assumptions that $\mathrm{D}_\sigma$ is locally homogeneous, and that $\nabla\,\mathrm{D}_\sigma$ is symmetric with spectral radius less than one, Romano et al. proved that $\mathrm{D}_\sigma = \mathrm{I} - \nabla\phi$. Yet the assumptions might be impractical for deep denoisers as reported by Reehorst & Schniter [54]. Instead of training a Gaussian denoiser $\mathrm{D}_\sigma$, Cohen et al. [16] parameterized $\phi$ with a neural network, $\mathrm{D}_\sigma = \nabla\phi$. Unfortunately, as verified by Salimans & Ho [61], and Hurault et al. [28], directly modeling $\phi$ with a neural network leads to poor performance. To tackle this, Hurault et al. [28] introduced the gradient step (GS) denoiser, where $\mathrm{D}_\sigma = \nabla\phi$, with $\phi(x) = \frac{1}{2}\|x\|^2 - g_\sigma(x)$, $g_\sigma(x) = \frac{1}{2}\|x - N_\sigma(x)\|^2$, where $N_\sigma$ is a neural network. Then $\mathrm{D}_\sigma = \mathrm{I} - \nabla g_\sigma = N_\sigma + \mathrm{J}_{N_\sigma}^{\mathrm{T}}\,(\mathrm{I} - N_\sigma)$, where $\mathrm{J}_{N_\sigma}$ is the Jacobian matrix of $N_\sigma$. $N_\sigma$ uses the light DRUNet architecture [73], and such $\mathrm{D}_\sigma$ is called GS-DRUNet. Following this, they then proposed Prox-DRUNet, which trains a GS-DRUNet with a non-expansive residual $\mathrm{I} - \mathrm{D}_\sigma$ via SR [29]. They proved that Prox-DRUNet acts as a proximal operator of some weakly convex prior $F$. The Prox-DRS algorithm is given by plugging $L\,\mathrm{D}_\sigma + (1 - L)\,\mathrm{I}$ as the denoiser into DRS. Prox-DRS converges for proper closed fidelity $G$ with $L \in [0, 0.5)$, making it applicable for Poisson inverse problems. However, this great work still requires a non-expansive residual, which alters the denoising performance, especially for large noises. To apply GS-DRUNet in Poisson inverse problems, Hurault et al. proposed to train a Bregman denoiser to remove Gamma noise [27]. Two convergent algorithms B-RED and B-PnP are derived. However, experimental results indicate that the Gamma denoiser struggles to remove large Gamma noise, thereby limiting the restoration performance.

**Motivations.** As discussed above, in order a get a proximal denoiser, it needs to be Lipschitz and conservative.

Typical Lipschitz conditions, such as non-expansiveness and residual non-expansiveness, are restrictive and lead to compromised performance. Weaker assumptions, like (strictly) pseudo-contractive conditions, can not guarantee a proximal denoiser. This limitation motivates us to explore a more suitable assumption that ensures the denoiser remains proximal while maintaining good denoising performance.

For the conservativeness, existing approaches typically construct an explicit potential function $\phi$. While this idea is intuitively appealing and convenient for optimization, it raises a question: Is there any alternative approach to promote the conservativeness, without requiring an explicit prior, by directly regularizing the denoiser, without significantly changing its network structure?

**Contributions.** To address the issues outlined above, this paper introduces a novel training strategy for learning a proximal denoiser. Leveraging the SR technique and the generalized Helmholtz decomposition, we propose to train a cocoercive conservative (CoCo) denoiser. CoCo denoiser proves to be a proximal operator of some implicit non-convex prior function. Cocoerciveness is a weaker constraint on the denoiser, compared to existing constraints like firm or residual non-expansiveness. As a result, the CoCo denoiser not only achieves superior denoising performance but also enhances PnP recovery in Poisson inverse problems. Overall, our main contributions are fourfold:

• We introduce a novel assumption, cocoerciveness, for the deep denoisers. Cocoerciveness is strictly weaker assumption than non-epansive type assumptions, and therefore, less restrictive for denoisers.
• We shed a new light on the conservative assumption, by studying the denoising geometry. Using the generalized Helmholtz decomposition, a denoiser is implicitly decomposed into a conservative part and a Hamiltonian part. We show intuitively and rigorously that an ideal denoiser should be conservative with no Hamiltonian part.
• We prove that a CoCo denoiser is a proximal operator of some implicit prior. An effective training

strategy is proposed to promote these properties.

• A Poisson inverse model with an implicit prior is derived. The global convergence of PnP methods with CoCo denoisers is established.

## 2 CoCo denoisers

In this section, we propose CoCo denoisers. First, we introduce the cocoercive assumption, and its spectral distribution on the complex plane. Next, by the generalized Helmholtz decomposition, we show intuitively that an ideal denoiser should be conservative, with no Hamiltonian part. Then, we prove that a CoCo denoiser $D_\sigma$ is the proximal operator of some weakly convex and smooth prior function $F : V \to \bar{\mathbb{R}} := \mathbb{R} \cup \{\infty\}$, $D_\sigma = \operatorname{Prox}_{\frac{F}{\beta}}$. Finally, a restoration model with an implicit weakly convex prior is given.

### 2.1 Cocoercive denoisers

Let $V = \mathbb{R}^n$ be a real Hilbert space[3] with inner product $\langle \cdot, \cdot \rangle$, and the induced norm $\|x\| = \sqrt{\langle x, x \rangle}$. Let $D : V \to V$ be an operator[4]. An operator $D : V \to V$ is said to be $\gamma$-cocoercive ($\gamma \in [0, \infty)$), if $\forall x, y \in V$, there holds:

$$\langle x - y, D(x) - D(y) \rangle \geq \gamma \| D(x) - D(y) \|^2. \tag{5}$$

Cocoercive operators are an important class of monotone operators. Many operators are cocoercive operators. Below are two typical cocoercive operators:

• D is firmly non-expansive: ($\forall x, y \in V$)

$$\langle x - y, D(x) - D(y) \rangle \geq \| D(x) - D(y) \|^2. \tag{6}$$

Such D is 1-cocoercive.

• D is residual non-expansive: ($\forall x, y \in V$)

$$\|(I - D) \circ (x) - (I - D) \circ (y)\| \leq \|x - y\|. \tag{7}$$

Such D is 0.5-cocoercive.

When $\gamma > 0$, cocoercive assumption make denoisers Lipschitz: by Cauchy-Schwarz inequality,

$$\langle x - y, D(x) - D(y) \rangle \leq \|x - y\| \| D(x) - D(y) \|. \tag{8}$$

Therefore, by (5), $\| D(x) - D(y) \| \leq \frac{1}{\gamma} \|x - y\|$, that is, D is $\frac{1}{\gamma}$-Lipschitz. When $\gamma < 0.5$, both the operator and its residual part are expansive. In general, a smaller $\gamma$ corresponds to less contraint, and therefore a more powerful denoiser.

**Spectral analysis on** D. Note that (5) can be equivalently transformed into the following inequality: ($\forall x, y \in V$)

$$\|(2\gamma D - I) \circ (x) - (2\gamma D - I) \circ (y)\| \leq \|x - y\|. \tag{9}$$

Based on the mean value theorem [18], (9) is transformed into:

$$\|2\gamma J(x) - I\|_* \leq 1, \forall x \in V, \tag{10}$$

where $J(x)$ is the Fréchet differential, that is the Jacobian matrix when $V$ is finite dimensional, of D at the point $x$, and $\| \cdot \|_*$ denotes the spectral norm. Since the spectral radius $\rho(\cdot)$ is always no larger than the spectral norm $\| \cdot \|_*$, we have that:

$$\rho(2\gamma J(x) - I) \leq 1, \forall x \in V. \tag{11}$$

---

[3]Please note that, although we limit $V$ to be finite dimensional for easy understanding, many theoretical analysis still hold in a general infinite dimensional real Hilbert space. To clarify, the spectrum set of an operator in a finite space is just the eigenvalue set. The transpose of a matrix in a finite dimensional space corresponds to the adjoint operator in an infinite space. $\| \cdot \|_*$ is the spectral norm in a finite space, and denotes the operator norm in an infinite space. The degradation operator $K$ in an infinite space is assumed to be a bounded linear operator, that is $K \in \mathcal{B}[V]$.

[4]An operator in general may not be single-valued. However, since the operator D in this paper is a denoiser, its output is unique given fixed input. Thus, we only consider the single-valued operator here.

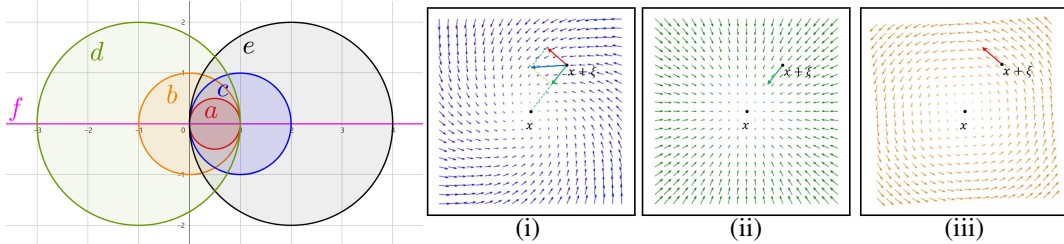

Figure 1: Left: Spectrum distributions of the Fréchet differential (Jacobian matrix) on the complex plane under different assumptions. (a) Firmly non-expansiveness, $\mathrm{Sp(J)} \subset \{z \in \mathbb{C} : |2z - 1| \leq 1\}$; (b) Non-expansiveness, $\mathrm{Sp(J)} \subset \{z \in \mathbb{C} : |z| \leq 1\}$; (c) Residual non-expansiveness, $\mathrm{Sp(J)} \subset \{z \in \mathbb{C} : |z - 1| \leq 1\}$; (d) $\frac{1}{2}$-strictly pseudo-contractiveness, $\mathrm{Sp(J)} \subset \{z \in \mathbb{C} : |z + 1| \leq 2\}$; (e) 0.25-cocoerciveness, $\mathrm{Sp(J)} \subset \{z \in \mathbb{C} : |0.5z - 1| \leq 1\}$; (f) Conservativeness, $\mathrm{Sp(J)} \subset \mathbb{R}$. In general, a larger region means less restrictive assumption. The spectrum of $\gamma$-CoCo denoisers ($\gamma = 0.25$) lies inside the interval $(f) \cap (e) = [0, 4]$. Spectrum outside $\mathbb{R}$ corresponds to the Hamiltonian part of the denoiser, and does not contribute to the denoising performance.
Right: A two-dimensional illustration of the Helmholtz decomposition of a denoiser $\mathrm{D}$. $x$ denotes the clean image point, $\xi$ denotes the Gaussian noise, and $x + \xi$ denotes the noisy image. The arrow "$\rightarrow$" represents the denoising direction. (i) Denoising field $\mathrm{D}$. (ii) Conservative field $\mathrm{D}_c$. (iii) Hamiltonian field $\mathrm{D}_h$.

Let $\mathrm{Sp}(\cdot)$ denote the spectrum set, that is, the eigenvalue set when $V$ is finte dimensional:

$$\mathrm{Sp(J)} := \{z \in \mathbb{C} : z\,\mathrm{I} - \mathrm{J} \text{ is not invertible}\}. \tag{12}$$

(11) implies $\mathrm{Sp(J}(x)) \subseteq \{z \in \mathbb{C} : |2\gamma z - 1| \leq 1\}$.

Lemma 2.1 gives an *equivalent* condition of cocoerciveness in (5)-(9), and thereby enables the training through SR.

**Lemma 2.1** (Proof in Appendix A.1). *Let* $\mathrm{D} : V \rightarrow V$, *and* $\mathrm{J} = \nabla \mathrm{D}$ *be its Fréchet differential. Then* $\mathrm{D}$ *is* $\gamma$-*cocoercive* ($\gamma \in [0, \infty)$)*, if and only if* $\|2\gamma\,\mathrm{J}(x) - \mathrm{I}\|_* \leq 1$ *for any* $x \in V$.

Since the spectral radius is no larger than the spectral norm, we are able to plot the spectral distribution on the complex plane $\mathbb{C}$ in Fig. 1. A larger region corresponds to a less restrictive constraint, and thus a better denoising performance. Figs. 1 (a), (c), (e) show that firm non-expansiveness, and residual non-expansiveness are special cases of cocoerciveness. Therefore, cocoeriveness is a weaker assumption for deep denoisers, and yields better denoising performance.

## 2.2 Conservative denoisers

We consider the conservative assumption intuitively. Let $\mathrm{D} \in \mathcal{C}^1[V]$ be a denoiser. $\mathrm{D}$ is said to be conservative, if there is a potential function $\phi : V \rightarrow \mathbb{R}$, such that $\mathrm{D} = \nabla\phi$. Geometrically, $\mathrm{D}$ is mapping from $V$ to $V$, thus can be viewed as a vector field.

By the generalized Helmholtz decomposition [8, 4], any vector field $\mathrm{D}$ can be decomposed into a conservative field $\mathrm{D}_c$, and a Hamiltonian field $\mathrm{D}_h$, such that $\mathrm{D}_c$ is curl-free and is the gradient of a potential function $\phi$, and $\mathrm{D}_h$ is divergence-free:

$$\mathrm{D} = \mathrm{D}_c + \mathrm{D}_h, \ \mathrm{D}_c = \nabla\phi, \ \mathrm{div}(\mathrm{D}_h) = 0. \tag{13}$$

However, the decomposition $\mathrm{D} = \mathrm{D}_c + \mathrm{D}_h$ is implicit. In order to characterize this decomposition, we consider the Jacobian matrix $\mathrm{J}$. The generalized Helmholtz decomposition can be rewritten as $\mathrm{J} = \mathrm{S} + \mathrm{A}$, where $\mathrm{S} = \frac{\mathrm{J} + \mathrm{J}^\top}{2}$ is symmetric, and $\mathrm{A} = \frac{\mathrm{J} - \mathrm{J}^\top}{2}$ is anti-symmetric. $\mathrm{S}$ and $\mathrm{A}$ correspond to $\mathrm{D}_c$ and $\mathrm{D}_h$ respectively: $\mathrm{S} = \nabla \mathrm{D}_c$, $\mathrm{A} = \nabla \mathrm{D}_h$.

In Fig. 1, we consider a two-dimensional case. $x \in V$ is a clean image, $\xi$ is the Gaussian noise with zero mean and standard derivation $\sigma$, $x + \xi$ is the noisy image. $\|x + \xi - x\| = \mathbb{E}[\|\xi\|] = 2\sigma$ is the distance between the clean image and the noisy image in $V$. A denoiser $\mathrm{D}$ is expected to reduce the distance as in Fig. 1 (i). That is $\|\mathrm{D}(x + \xi) - x\| < \|x + \xi - x\|$. The arrow "$\rightarrow$" in Fig. 1 denotes the denoising direction vector $\eta$, $\eta = \mathrm{D}(x + \xi) - (x + \xi)$. $\eta$ can be decomposed into $\eta_c$ and $\eta_h$, see

Figs. 1 (ii)-(iii). $\eta_c$ points to the clean image, and thus represents the correct denoising direction. $\eta_h$ circles around $x$, and $D_h$ cannot remove noises.

In a general real Hilbert space, given a point $x$, $\forall y \in V$, define the Hamiltonian functional $\mathcal{H}(y) = \frac{1}{2}\| D(y) - x \|^2$. When D is a Hamiltonian field, S $= 0$. Then, the vector $D(y) - x$ preserves the level set of $\mathcal{H}$, because:

$$\langle D(y) - x, \nabla_y \mathcal{H}(y) \rangle = \langle D(y) - x, A^\top (D(y) - x) \rangle = 0.$$

Note that when $x$ is a clean image, $\mathcal{H}$ is a typical loss function for a denoiser. Therefore, a Hamiltonian field does not contribute to the denoising. Thus we penalize this useless part $D_h$ by penalizing A. The following relations are useful: D is conservative $\iff D_h = 0 \iff A = 0 \iff \| J - J^\top \|_* = 0$.

In order to penalize $D_h$, we only need to minimize $\| J - J^\top \|_*$. Spectrally speaking, when $J = J^\top$ is self-adjoint, $\text{Sp}(J) \subset \mathbb{R}$, see Fig. 1 (f).

## 2.3 Characterization on CoCo denoisers

In the context of PnP, the denoiser $D_\sigma$ is expected to be proximal. Building upon prior findings by [22] (see Appendix A.2), we present the following Theorem 2.2. A more general characterization on CoCo denoisers is given in Theorem A.5 in Appendix A.3.

**Theorem 2.2** (Proof in Appendix A.4). *Let* $D_\sigma \in \mathcal{C}^1[V]$. $\beta = \frac{1}{\sigma^2}$. $D_\sigma$ *satisfies that:*
- $D_\sigma$ *is conservative;*
- $D_\sigma$ *is $\gamma$-cocoercive with $\gamma \in (0,1)$.*
*Let* $D_\sigma^t = t\,D_\sigma +(1-t)\,I$, $t \in [0,1)$. *It holds that:*
- *there exists a $r$-weakly convex function* $F : V \to \bar{\mathbb{R}}$, $r(t) = \beta\frac{t-\gamma t}{t+\gamma-\gamma t}$, *such that* $D_\sigma^t(x) \in \text{Prox}_{\frac{F}{\beta}}(x), \forall x \in V$;
- $\partial F$ *is $L$-Lipschitz, where*

$$L(t) = \begin{cases} \beta\frac{t}{1-t} \geq r(t), & \text{if } t \geq \frac{1-2\gamma}{2-2\gamma} \text{ and } t \in [0,1) \text{ (Case 1);} \\ r(t) = \beta\frac{t-\gamma t}{t+\gamma-\gamma t}, & \text{if } t \leq \frac{1-2\gamma}{2-2\gamma} \text{ and } t \in [0,1) \text{ (Case 2).} \end{cases} \tag{14}$$

*Remark* 2.3. $D_\sigma^t$ is an averaged version of $D_\sigma$. As reported by [29], when $D_\sigma$ is residual non-expansive, PnP with $D_\sigma^t$ out-performs PnP with $D_\sigma$ in many inverse problems.

*Remark* 2.4. If $\gamma = 0.5$, $D_\sigma$ is residual non-expansive, $D_\sigma^t$ is then residual $t$-Lipschitz, and $\beta^{-1}F$ is $\frac{t}{1+t}$-weakly convex, with $\beta^{-1}\partial F$ being $\frac{t}{1-t}$-Lipschitz. This special case recovers the result by [29].

*Remark* 2.5. By Theorem 2.2, we know that $D_\sigma^t$ is a proximal operator of a weakly convex function $\frac{F}{\beta}$. We arrive at the following implicit non-convex restoration model:

$$\hat{u} \in \arg\min_{u \in V} F(u) + G(u; f), \ G(u; f) = \lambda\langle \mathbf{1}, Ku - f \log Ku\rangle. \tag{15}$$

## 3 Training strategy

Let $\theta$ be the parameter weights in $D_\sigma$ to be optimized, $\pi$ be the distribution of the training set of clean images, and $[\sigma_{\min}, \sigma_{\max}]$ be the interval of the noise level.

Based on Lemma 2.1 and the pioneer works by [66, 51], we encourage the cocoerciveness by the SR technique. The spectral regularization term $L_s$ takes the form of:

$$L_s(\theta) = \mathbb{E}_{x,\sigma,\xi} \max\{\|2\gamma J - I\|_*, 1 - \epsilon\}, \tag{16}$$

where $\theta$ is the network weights, $\epsilon \in (0,1)$ is a parameter that controls the constraint. $x \sim \pi, \sigma \sim U[\sigma_{\min}, \sigma_{\max}], \xi \sim \mathcal{N}(0, \sigma^2 I)$. $J = J(x + \xi) = \nabla D_\sigma(x + \xi; \theta)$.

To encourage a conservative denoiser, we propose the Hamiltonian regularization term $L_h$:

$$L_h(\theta) = \mathbb{E}_{x,\sigma,\xi}\| J - J^\top \|_*. \tag{17}$$

Combining (17) and (16), the overall loss function of $D_\sigma$ is

$$Loss(\theta) = \mathbb{E}\| D_\sigma(x + \xi; \theta) - x\|_1 + \alpha_1 L_h + \alpha_2 L_s. \tag{18}$$

$\alpha_1, \alpha_2 > 0$ are balancing parameters. The first term in (97) ensures that $D_\sigma$ is a Gaussian denoiser, the second term makes $D_\sigma$ conservative, and the third term results in a $\gamma$-cocoercive denoiser. By minimizing $Loss(\theta)$, a properly regularized CoCo denoiser is obtained. Detailed calculation of $\|(2\gamma J - I)\|_*$ and $\| J - J^T \|_*$ is given in Appendix A.5.

## 4    PnP methods with CoCo denoisers

In order to solve the implicit nonconvex restoration model in (15), we first consider PnP-ADMM with CoCo denoisers, termed CoCo-ADMM. Then, we replace the fidelity term $G$ in (15) by its Moreau envelope, and derive the proximal envelope gradient descent method (PEGD) with CoCo denoisers (CoCo-PEGD). The convergence are provided.

**CoCo-ADMM.** We first consider CoCo-ADMM iterations:

$$
\begin{aligned}
u^{k+1} &= \mathrm{Prox}_{\frac{G}{\beta}}(v^k - b^k), \\
v^{k+1} &= D_\sigma^t(u^{k+1} + b^k), \\
b^{k+1} &= b^k + u^{k+1} - v^{k+1},
\end{aligned}
\tag{19}
$$

where $D_\sigma^t$ is defined in Theorem 2.2, and $\beta = \frac{1}{\sigma^2}$. The PnP-ADMM algorithm in (19) with a $\gamma$-CoCo denoiser is referred to as $\gamma$-CoCo-ADMM, or CoCo-ADMM for short.

When the denoiser $D_\sigma \in \mathcal{C}^1[V]$ is a CoCo denoiser satisfying the conditions in Theorem 2.2, and $F$ verifies the Kurdyka-Lojasiewicz (KL) property [1, 19], the global convergence of PnP-ADMM in (19) can be established as follows.

**Theorem 4.1** (Proof in Appendix A.6). *Let $F : V \to \bar{\mathbb{R}}$ be a coercive weakly convex KL function in Theorem 2.2 such that $D_\sigma^t \in \mathrm{Prox}_{\frac{F}{\beta}}$. $G : V \to \bar{\mathbb{R}}$ is lower semi-continuous and convex. $\gamma \in (0, 1)$. $t \in [0, t_0)$, where $t_0 = t_0(\gamma)$ is the positive root of the equation*

$$
(2 - 2\gamma)t^3 + \gamma t^2 + 2\gamma t - \gamma = 0.
\tag{20}
$$

*Then, the sequence $\{(u^k, v^k, b^k)\}$ generated by (19) converges globally to a point $(u^*, v^*, b^*)$, and that $u^* = v^*$ is a stationary point of the model (15).*

*Remark* 4.2. The KL property has been widely used to study the convergence of optimization algorithms in the nonconvex and nonsmooth setting [1]. Many functions, in particular all the real semi-algebraic functions, satisfy this property.

*Remark* 4.3. We consider several special cases of Theorem 4.1. Let $D_\sigma$ be conservative. When $\gamma = 0.5$, $D_\sigma$ is residual non-expansive, and $t_0 \approx 0.3761$. This recovers the case by [27]; when $\gamma = 0.25$, both $D_\sigma$ and its residual $I - D_\sigma$ can be expansive, but $0.5 D_\sigma - I$ is non-expansive. In this case, $t_0 \approx 0.3333$. When $\gamma = 1$, $D_\sigma$ is firmly non-expansive and is a proximal operator of some CCP function by Moreau's theorem. In this case, $t_0$ is no more needed to ensure the convergence, since this is the standard case of ADMM. In experiments, we use $D_\sigma(u^K + b^K)$ as the final output.

**CoCo-PEGD.** Now we consider the PGD algorithm with a CoCo denoiser. The standard PGD takes the form of:

$$
u^{k+1} = \mathrm{Prox}_{\frac{F}{\beta}}\left(u^k - \beta^{-1}\nabla G(u^k)\right).
\tag{21}
$$

When $F$ is $r$-weakly convex, $\nabla G$ is $L_G$-Lipschitz, and $\beta^{-1} < \max\{\frac{2}{L_G + r}, \frac{1}{L_G}\}$, the iteration (21) converges, see [2, 27].

However, in the Poisson inverse problems, $G$ is not smooth, because $\nabla G$ is not Lipschitz. In order to apply the PGD algorithm, we smooth the fidelity $G$ in (15) and (21) by replacing it with its Moreau envelope ${}^\alpha G$ ($\alpha \in (0, \infty)$): ${}^\alpha G(u) := \min_v G(v) + \frac{1}{2\alpha}\|v - u\|^2$.

Since $G$ is CCP, ${}^\alpha G$ is differentiable [7]: $\nabla^\alpha G(u) = \frac{1}{\alpha}(I - \mathrm{Prox}_{\alpha G})$, and that $\nabla^\alpha G$ is $\frac{1}{\alpha}$-Lipschitz. Since there are already a step parameter $\beta$ and a balancing parameter $\lambda$ in $G$, throughout this paper, we set $\alpha = 1$. Now the iteration becomes:

$$
u^{k+1} = \mathrm{Prox}_{\frac{F}{\beta}}(u^k - \beta^{-1}\nabla^1 G(u^k)).
\tag{22}
$$

In (22), $u$ alternates between a proximal operator $\mathrm{Prox}_{\frac{F}{\beta}}$, and a gradient descent step on the envelope of $G$. (22) is referred to as the proximal envelope gradient descent method (PEGD). Similar to

CoCo-ADMM (19), we replace $\mathrm{Prox}_{\frac{F}{\beta}}$ with $\mathrm{D}_\sigma^t$, and arrive at CoCo-PEGD:

$$u^{k+1} = \mathrm{D}_\sigma^t(u^k - \beta^{-1}\nabla^1 G(u^k)). \tag{23}$$

By Theorem 2.2, $\mathrm{D}_\sigma^t = \mathrm{Prox}_{\frac{F}{\beta}}$, with $F$ being $r$-weakly convex. Then by the Theorem 1 in [27], CoCo-PEGD converges when $\beta^{-1} < \max\{\frac{2}{1+r}, 1\}$. This convergence result is summarized in Theorem 4.4. By Theorem 4.4, when $\beta > 1$, CoCo-PEGD converges.

**Theorem 4.4** (Proof in Appendix A.7). *Let $F : V \to \bar{\mathbb{R}}$ be a coercive $r$-weakly convex KL function in Theorem 2.2 such that $\mathrm{D}_\sigma^t = \mathrm{Prox}_{\frac{F}{\beta}}$. $G : V \to \bar{\mathbb{R}}$ is CCP. $\gamma \in [0.25, 1]$. $t \in (0, 1]$. $0 < \beta^{-1} < \max\{\frac{2}{1+r}, 1\}$. Then the sequence $\{u^k\}$ generated in (23) converges to a stationary point of:*

$$\hat{u} \in \arg\min_{u \in V} F(u) + {}^1G(u; f). \tag{24}$$

## 5 Experiments

All the experiments are conducted under Linux system, Python 3.8.12 and Pytorch 1.10.2 with a RTX 3090 GPU.

**Training details.** For $\mathrm{D}_\sigma$, we select DRUNet [73], which combines a residual learning [24] and UNet architecture [58]. DRUNet takes both the noisy image and the noise level $\sigma$ as input, making it convenient for PnP image restoration.

To train a CoCo denoiser, we collect 800 images from the DIV2K dataset [30] as the training set and used a batch size of 32 and patch size 128. We add Gaussian noise with randomly generated standard deviation values in the range of $[\sigma_{\min}, \sigma_{\max}] = [0, 50]$ to the clean image. Adam optimizer is applied to train the model with learning rate $lr = 10^{-4}$. We set $\alpha_1 = 1, \alpha_2 = 0.01$, and $\epsilon = 0.1$ to ensure the regularity conditions. To accurately evaluate the spectral norms $\|\mathrm{J}(x) - \mathrm{J}^\top(x)\|_*$ and $\|(2\gamma\mathrm{J} - \mathrm{I})(x)\|_*$, we use the power iterative method [21] with 30 iterations to ensure the convergence.

**Denoising performance.** We evaluate the Gaussian denoising performances of the proposed denoiser CoCo-DRUNet, strictly pseudo-contractive DRUNet (SPC-DRUNet) [71], resolvent of a maximally monotone operator (RMMO) [51] which is firmly non-expansive, GS-DRUNet [28], Prox-DRUNet [29], the standard DRUNet, FFDNet [75], and DnCNN [74].

The PSNR values are given in Table 1. A $\gamma$-cocoercive conservative denoiser is referred to as $\gamma$-CoCo-DRUNet. We see in Table 1 that compared with DnCNN, FFDNet, and the regularized denoisers, 0.25-CoCo-DRUNet has competitive denoising performance. This is because that: cocoercive and conservative properties are less restrictive for deep denoisers; denoiser is trained with a different loss function. Both 0.50-CoCo-DRUNet and Prox-DRUNet are conservative with non-expansive residual, and therefore share a similar denoising performance.

Table 1: Left: Average denoising PSNR performance of different denoisers on CBSD68, for various noise levels $\sigma$; Right: Mean symmetry error $\|\mathrm{J} - \mathrm{J}^\top\|_*$ ($N = 1$) and maximal values of the norm $\|2\gamma\mathrm{J} - \mathrm{I}\|_*$ ($N = 30$) for various noise levels $\sigma$ and $\gamma = 0.50, 0.25$.

|  | $\sigma = 15$ | $\sigma = 25$ | $\sigma = 40$ |
|---|---|---|---|
| FFDNet | 33.86 | 31.18 | 28.81 |
| DnCNN | 33.88 | 31.20 | 28.89 |
| DRUNet | **34.14** | **31.54** | **29.33** |
| RMMO | 32.21 | 29.99 | 27.87 |
| GS-DRUNet | 33.56 | 31.01 | 28.81 |
| Prox-DRUNet | 33.18 | 30.60 | 28.38 |
| SPC-DRUNet | 33.90 | 31.29 | 29.10 |
| 0.50-CoCo-DRUNet | 33.38 | 30.65 | 28.25 |
| 0.25-CoCo-DRUNet | **34.00** | **31.38** | **29.16** |

|  | 15 | 25 | 40 | Norms |
|---|---|---|---|---|
| DRUNet | 4.1e+0 | 4.2e+1 | 1.8e+1 | $\|\mathrm{J} - \mathrm{J}^\top\|_*$ |
| 0.50-CoCo-DRUNet | 8.6e-3 | 2.5e-4 | 7.1e-4 | $\|\mathrm{J} - \mathrm{J}^\top\|_*$ |
| 0.25-CoCo-DRUNet | 3.1e-4 | 1.8e-4 | 3.9e-4 | $\|\mathrm{J} - \mathrm{J}^\top\|_*$ |
| DRUNet | 3.285 | 4.343 | 6.283 | $\|\mathrm{J} - \mathrm{I}\|_*$ |
| 0.50-CoCo-DRUNet | 0.994 | 0.992 | 0.972 | $\|\mathrm{J} - \mathrm{I}\|_*$ |
| 0.25-CoCo-DRUNet | 0.986 | 0.969 | 0.982 | $\|0.5\,\mathrm{J} - \mathrm{I}\|_*$ |

**Assumption validations.** In the experiments, the symmetric Jacobian and non-expansive residual are softly constrained by the loss function (97) with trade-off parameters $\alpha_1, \alpha_2$. We validate the conditions in Table 1. We use the symmetry error $\|\mathrm{J}(x) - \mathrm{J}^\top(x)\|_*$ over 100 different patches with $N = 1$ as in Algorithm 1 for better demonstration. A smaller error denotes a higher Jacobian symmetry. The cocoerciveness is validated by calculating $\|2\gamma\mathrm{J}(x) - x\|_*$. If $\|2\gamma\mathrm{J}(x) - x\|_* \le 1$, the denoiser is $\gamma$-cocoercive.

As shown in Table 1, DRUNet without regularization terms has an expansive residual part. The regularized CoCo-DRUNet is cocoercive. Besides, the proposed Hamiltonian regularization term indeed encourages a symmetric Jacobian: in Table 1, we see that Hamiltonian regularization reduces the symmetry error. It validates the effectiveness of the proposed training strategy. Ablation study on the parameters $\alpha_1$ and $\alpha_2$ in (97) are provided on Appendix A.16.

**PnP restoration.** We apply CoCo-ADMM and CoCo-PEGD to multiple Poisson inverse problems including: **photon limited deconvolution**, **single photon imaging** in a real-world low-light setting in Appendix A.10, **low-dose CT reconstruction** tasks in Appendix A.12, **Poisson denoising** in Appendix A.13. **Computational time**, **performances under extreme conditions**, and **blind deblur and denoise results**, are reported in Appendices A.14, A.15, A.17, and A.18 respectively.

When $K$ in (2) is not the identity matrix, in general, $\mathrm{Prox}_G$ has no closed-form solution. Since $G$ is convex, we use ADMM to solve $\mathrm{Prox}_G$ efficiently. For details, please refer to Appendix A.8.

We choose $\gamma = 0.25$ since 0.25-CoCo-DRUNet has a satisfactory performance. According to Theorems 4.1-4.4, CoCo-ADMM and CoCo-PEGD are guaranteed to converge with $t < 0.3333$. For both CoCo-ADMM and PEGD, we need to calculate the proximal operator $\mathrm{Prox}_{\frac{G}{\beta}}$. We detail the calculations for each task in Appendix A.8. For each method, we fine tune the parameters to achieve the best quantitive PSNR values. The proposed methods are initialized with the observed image, that is $u^0 = v^0 = f, b^0 = 0$.

Table 2: Left: Average deconvolution PSNR and SSIM performance by different methods on CBSD68 with Levin's 8 kernels and Poisson noises with peak value $p = 50$ and $p = 100$. Right: Average low-dose CT reconstruction PSNR and SSIM performance by different methods on Mayo's dataset with Poisson noises with peak value $p = 500$, and $p = 100$.

| | $p = 100$ | | $p = 50$ | |
|---|---|---|---|---|
| | PSNR | SSIM | PSNR | SSIM |
| DPIR | 26.51 | **0.7419** | 25.38 | 0.6910 |
| RMMO-DRS | 25.94 | 0.7019 | 25.10 | 0.6546 |
| Prox-DRS | 25.68 | 0.6764 | 25.21 | 0.6572 |
| DPS | 23.65 | 0.6062 | 23.10 | 0.5816 |
| DiffPIR | 24.82 | 0.6429 | 24.08 | 0.6074 |
| SNORE | 26.33 | 0.7158 | 25.21 | 0.6719 |
| PnPI-HQS | 26.42 | 0.7158 | 25.61 | 0.6956 |
| B-RED | 24.09 | 0.6794 | 23.78 | 0.6603 |
| CoCo-ADMM | **26.89** | 0.7358 | **26.00** | **0.7026** |
| CoCo-PEGD | 26.79 | 0.7323 | 25.90 | 0.6958 |

| | $p = 500$ | | $p = 100$ | |
|---|---|---|---|---|
| | PSNR | SSIM | PSNR | SSIM |
| FBP | 28.76 | 0.5212 | 24.10 | 0.2974 |
| PWLS-TGV | 33.16 | 0.8461 | 30.43 | 0.7750 |
| PWLS-CSCGR | 35.45 | 0.8909 | 33.78 | 0.8534 |
| UNet | 36.93 | 0.9139 | 35.19 | 0.8875 |
| WNet | 37.12 | 0.9266 | 35.98 | 0.9130 |
| CoCo-ADMM | 37.63 | **0.9403** | **36.68** | **0.9194** |
| CoCo-PEGD | **37.72** | 0.9391 | 36.43 | 0.9159 |

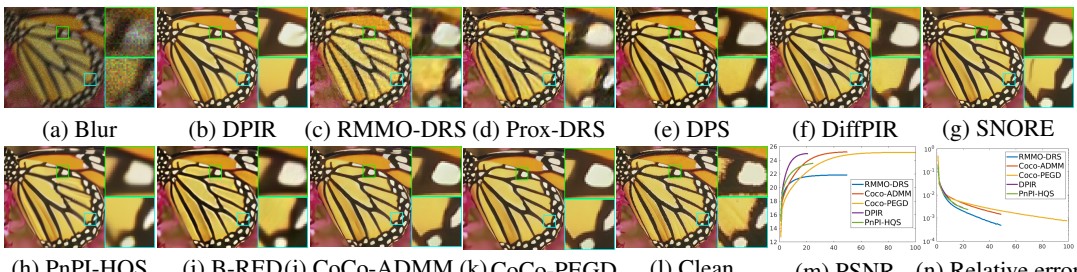

(a) Blur    (b) DPIR    (c) RMMO-DRS (d) Prox-DRS    (e) DPS    (f) DiffPIR    (g) SNORE

(h) PnPI-HQS    (i) B-RED (j) CoCo-ADMM (k) CoCo-PEGD    (l) Clean    (m) PSNR    (n) Relative error

Figure 2: Deconvolution results by different methods on the image 'Butterfly' from Set3c with kernel 2 and $p = 50$ Poisson noises. (a) Blur image. (b) DPIR, PSNR=25.00dB. (c) RMMO-DRS, PSNR=21.86dB. (d) Prox-DRS, PSNR=22.57dB. (e) DPS, PSNR=21.47dB. (f) DiffPIR, PSNR=22.66dB. (g) SNORE, PSNR=25.14dB. (h) PnPI-HQS, PSNR=23.56dB. (i) B-RED, PSNR=23.34dB. (j) CoCo-ADMM, PSNR=25.25dB. (k) CoCo-PEGD, PSNR=25.17dB. (l) Clean image. (m) PSNR curves. (n) Relative error curves. $x$-axis denotes the iteration number.

**Photon limited deconvolution.** In this task, $K$ in (15) is the blur kernel. We use 8 real-world camera shake kernels by [39], see Fig. 4 in Appendix A.9.

We compare our methods with some close related PnP methods, including DPIR [73], which applies PnP-HQS method with decreasing step size. Please note that DPIR is not guaranteed to converge;

RMMO-DRS [66], which uses the DRS method with a firmly non-expansive denoiser RMMO; B-RED [27], which uses the gradient descent method with a Bregman denoiser; PnPI-HQS [71], which uses the Ishikawa HQS method with a strictly pseudo-contractive denoiser; SNORE [55], which proposes the novel stochastic denoising regularization by iteratively adding Gaussian noises. We also compare two state-of-the-art diffusion-based methods: DPS [15] and DiffPIR [77]. The average PSNR and SSIM values on CBSD68 are summarized in Table 2.

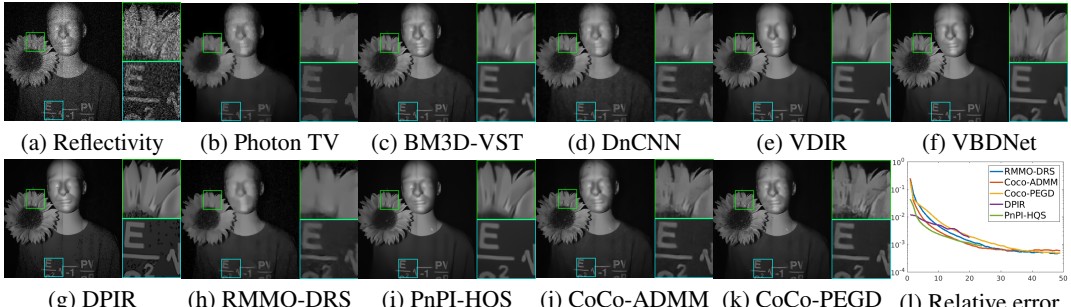

| (a) Reflectivity | (b) Photon TV | (c) BM3D-VST | (d) DnCNN | (e) VDIR | (f) VBDNet |

| (g) DPIR | (h) RMMO-DRS | (i) PnPI-HQS | (j) CoCo-ADMM | (k) CoCo-PEGD | (l) Relative error |

Figure 3: Single photon imaging results in a real-world low light setting by different methods.

We show the visual results in Fig. 2. In Fig. 2 (a), the image 'Butterfly' is severely degraded. In this setting, many methods fail to recover the clear edges, see Figs. 2 (c)-(i). Compared with DPIR and SNORE, the enlarged parts by CoCo-ADMM and CoCo-PEGD are closer to the potential clean image, see Figs. 2 (b), (e), (h), (i).

## 6    Conclusion and Limitation

This paper introduces a novel cocoercive conservative assumption on the denoiser. Cocoerciveness is weaker than the existing assumptions, and is less restrictive for a deep denoiser. Conservativeness is analyzed geometrically by the generalized Helmholtz decomposition on the Fréchet differential. We propose a novel training strategy that incorporates a Hamiltonian regularization term and a spectral regularization term, which encourages a cocoercive conservative (CoCo) Gaussian denoiser. Theoretically, CoCo denoiser is proved to be a proximal operator of an implicit weakly convex prior function. The global convergence results of PnP methods to a stationary point are given. The results can be naturally generalized to other inverse problems with a convex fidelity term and an implicit weakly convex prior term. Extensive experimental results demonstrate that the proposed CoCo-PnP methods achieve competitive performance in terms of both visual quality and quantitative measures.

The main limitation of this paper is that the proposed CoCo denoiser exhibits a slightly worse Gaussian denoising performance compared to the non-regularized denoiser, see Table 6. This is because the proposed regularized denoiser sacrifices a little denoising performance in order to achieve guaranteed theoretical results as stated in Theorems 2.2-4.4. Another limitation is that the proposed learning strategy can only encourage the desired mathematical properties, not enforce them. In experiments, we do observe that sometimes, even after a long time spectral regularization training, the trained denoiser still violates the cocoercive property on some particular images. For conservativity, we find that when $N = 30$, the symmetry error compared with standard DRUNet is smaller, but not significantly smaller. How to enforce such properties without greatly compromising the denoising performance is still an open problem. Since the paper mainly focuses on the theoretical analysis of PnP methods, we will address these limitations in future works.

## Acknowledgments

Fang Li is supported by National Key R&D Program of China (Nos. 2021YFA1000302), Fundamental Research Funds for the Central Universities, Natural Science Foundation of Shanghai (No. 22ZR1419500), Science and Technology Commission of Shanghai Municipality (No. 22DZ2229014), and Natural Science Foundation of Chongqing, China (No. CSTB2023NSCQ-MSX0276). Tieyong Zeng is supported in part by BNBU Research Grant (No. UICR0100031, UICR0700108-25 and UICR0900003) at Beijing Normal-Hong Kong Baptist University, Zhuhai, PR China.

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

# A  Technical Appendices and Supplementary Material

## Contents

## A.1 Proof of Lemma 2.1

In order to prove Lemma 2.1, we will mainly use Proposition 3.1 by [6]:

**Proposition A.1** (Proposition 3.1 by [6])**.** *Let $C$ be a nonempty open convex subset of $V$, let $\mathcal{B}$ be a real Banach space, and let $\mathrm{D} : C \to \mathcal{B}$ be continuously Fréchet differentiable on $C$. Then $\mathrm{D}$ is non-expansive if and only if $\| \mathrm{J}(x) \|_* \leq 1$, $\forall x \in C$, $\mathrm{J}(x) := \nabla \mathrm{D}(x)$.*

Now we prove Lemma 2.1.

*Proof.* In Lemma 2.1, we let $C = \mathcal{B} = V$, thus $C$ is open convex, and $\mathcal{B}$ is a real Banach space. In order to prove Lemma 2.1, we only need to prove that $\mathrm{D}$ is $\gamma$-cocoercive ($\gamma \in [0, \infty)$), if and only if $\forall x, y \in V$

$$\| (2\gamma \mathrm{D} - \mathrm{I}) \circ (x) - (2\gamma \mathrm{D} - \mathrm{I}) \circ (y) \| \leq \| x - y \|. \tag{25}$$

Note that (25) is equivalent to

$$4\gamma^2 \| \mathrm{D}(x) - \mathrm{D}(y) \|^2 + \| x - y \|^2 - 4\gamma \langle x - y, \mathrm{D}(x) - \mathrm{D}(y) \rangle \leq \| x - y \|^2, \tag{26}$$

and equivalent to

$$\gamma \| \mathrm{D}(x) - \mathrm{D}(y) \|^2 \leq \langle x - y, \mathrm{D}(x) - \mathrm{D}(y) \rangle, \tag{27}$$

which means $\mathrm{D}$ is $\gamma$-cocoercive. $\square$

## A.2 Characterizations on the denoiser

The first characterization of proximal operators of convex, closed, and proper functions is due to Moreau's theorem [47].

**Theorem A.2** ([47])**.** *A map $\mathrm{D} : V \to V$ is a proximal operator of a proper, closed, convex function $F : V \to \mathbb{R} \cup \{+\infty\}$, if and only if the it holds that:*

- *there exists a convex, closed function $\psi$ such that $\mathrm{D}(x) \in \partial \psi(x), \forall x \in V$;*

- $\mathrm{D}$ *is nonexpansive, i.e.,*

$$\| \mathrm{D}(x) - \mathrm{D}(y) \| \leq \| x - y \|, \forall x, y \in V. \tag{28}$$

[22] generalized Moreau's theorem to the proximal operators of potentially nonconvex functions.

**Theorem A.3** ([22])**.** *Let $\mathrm{D} : V \to V$ and $L > 0$. The following are equivalent:*

- *there is $F : V \to \mathbb{R} \cup \{+\infty\}$ such that $\mathrm{D}(x) \in \mathrm{Prox}_F(x), \forall x \in V$, and $x \mapsto F(x) + \left(1 - \frac{1}{L}\right) \frac{\|x\|^2}{2}$ is closed and convex;*

- $\mathrm{D}$ *is $L$-Lipschitz, and that there exists a closed, convex function $\psi$ such that $\mathrm{D}(x) \in \partial \psi(x), \forall x \in V$.*

When $\mathrm{D} \in \mathcal{C}^1[V]$, Gribonval and Nikolova also provided the following characterization.

**Theorem A.4** ([22])**.** *Let $\mathrm{D} : V \to V$, and $\mathrm{D} \in \mathcal{C}^1[V]$. The following properties are equivalent:*

- $\mathrm{D}$ *is a proximal operator of a function $F : V \to \mathbb{R} \cup \{+\infty\}$;*

- *there exists a convex $\mathcal{C}^2[V]$ function $\psi$ such that $\mathrm{D}(x) = \nabla \psi(x), \forall x \in V$;*

- *the differential $\mathrm{J}(x) = \nabla \mathrm{D}(x)$ is symmetric positive semi-definite for all $x \in V$.*

## A.3 More general characterization on the CoCo denoiser

The following theorem is a tight use of Theorem A.4.

**Theorem A.5.** *Let $\mathrm{D}_\sigma \in \mathcal{C}^1[V]$. $\beta = \frac{1}{\sigma^2}$. $\mathrm{D}_\sigma$ satisfies that:*
$\bullet$ $\mathrm{D}_\sigma$ *is conservative;*
$\bullet$ $\mathrm{D}_\sigma$ *is $\gamma$-cocoercive with $\gamma \in (0, \infty)$.*
*Then, there exists a function $F : V \to \bar{\mathbb{R}}$, such that $F$ is $r$-weakly convex, where $r = \beta(1 - \gamma)$, and that $\mathrm{D}_\sigma(x) \in \mathrm{Prox}_{\frac{F}{\beta}}(x), \forall x \in V$.*

Note that here is a little abuse of notation here: when $\gamma > 1$, $r < 0$. "$r$-weakly convexity with a negative $r$" actually means "$(-r)$-strongly convexity".

*Proof.* When $D_\sigma$ is $\gamma$-cocoercive with $\gamma > 0$, by Lemma 2.1, $\forall x \in V$, $\|2\gamma J(x) - I\|_* \leq 1$. Therefore, $J(x)$ is positive semi-definite for any $x \in V$. Since $D_\sigma$ is conservative, it has no Hamiltonian part, $J(x) = J^T(x), \forall x \in V$. Therefore, $J(x)$ is symmetric semi-positive for any $x \in V$. By Poincaré's lemma (see Theorem 6.6.3 in [20]), there exists a convex $C^2[V]$ function $\psi$ such that $D_\sigma(x) = \nabla\psi(x), \forall x \in V$.

By Theorem A.4, there exists a function $F : V \to \bar{\mathbb{R}}$, such that $D_\sigma \in \text{Prox}_{\frac{F}{\beta}}$, where we let $\beta = \frac{1}{\sigma^2}$ for convenience. Now we prove that $F$ is weakly convex. Recall the resolvent form of a proximal operator: $\text{Prox}_{\frac{F}{\beta}} = (I + \frac{1}{\beta}\partial F)^{-1}$. Given any $x, y \in V$, choose arbitrary $u, v \in V$, $u = D_\sigma(x) \in (I + \frac{1}{\beta}\partial F)^{-1}(x), v = D_\sigma(y) \in (I + \frac{1}{\beta}\partial F)^{-1}(y)$, then

$$\beta(x - u) \in \partial F(u), \beta(y - v) \in \partial F(v). \tag{29}$$

Since $D_\sigma$ is $\gamma$-cocoercive, by definition 5,

$$\langle x - y, D_\sigma(x) - D_\sigma(y)\rangle \geq \gamma\|D_\sigma(x) - D_\sigma(y)\|^2. \tag{30}$$

By substituting $u, v$ in (30), we have

$$\begin{aligned}
\langle x - y, u - v\rangle &\geq \gamma\|u - v\|^2 \\
\langle (x - u) - (y - v) + (u - v), u - v\rangle &\geq \gamma\|u - v\|^2 \\
\langle \beta(x - u) - \beta(y - v) + \beta(u - v), u - v\rangle &\geq \beta\gamma\|u - v\|^2 \\
\langle \beta(x - u) - \beta(y - v), u - v\rangle &\geq \beta(\gamma - 1)\|u - v\|^2 \\
\langle \beta(x - u) - \beta(y - v), u - v\rangle + \beta(1 - \gamma)\|u - v\|^2 &\geq 0.
\end{aligned} \tag{31}$$

Recall (29), we know that $F$ is $r$-weakly convex with $r = \beta(1 - \gamma)$. $\qquad\square$

## A.4 Proof of Theorem 2.2

Please note that, when $\gamma \geq 1$, $D_\sigma$ is firmly non-expansive and conservative, and therefore is a proximal operator of some convex function. In this case, $r(t)$ could be negative: the "negative weakly convexity" actually means convexity. Since a convex implicit prior function $F$ is out of interest in this paper, Theorem 2.2 focuses on the weakly convex case, that is, $\gamma \in (0, 1)$.

*Proof.* Since $D_\sigma$ is $\gamma$-cocoercive with $\gamma > 0$, $D_\sigma^t$ is naturally $\frac{\gamma}{t + (1-t)\gamma}$-cocoercive: $\forall x, y \in V$, we have

$$\begin{aligned}
\langle D_\sigma(x) - D_\sigma(y), x - y\rangle &\geq \gamma\|D_\sigma(x) - D_\sigma(y)\|^2 \\
\langle \tfrac{1}{t}(D_\sigma^t - (1-t)I) \circ (x) - \tfrac{1}{t}(D_\sigma^t - (1-t)I) \circ (y), x - y\rangle & \\
\geq \quad \gamma\|\tfrac{1}{t}(D_\sigma^t - (1-t)I) \circ (x) - \tfrac{1}{t}(D_\sigma^t - (1-t)I) \circ (y)\|^2 & \\
t\langle D_\sigma^t(x) - D_\sigma^t(y), x - y\rangle - t(1-t)\|x - y\|^2 &\geq \gamma\|D_\sigma^t(x) - D_\sigma^t(y) - (1-t)(x - y)\|^2.
\end{aligned} \tag{32}$$

Denote $a = D_\sigma^t(x) - D_\sigma^t(y), b = x - y$ for convenience. Then we have

$$\begin{aligned}
t\langle a, b\rangle - t(1-t)\|b\|^2 &\geq \gamma\|a\|^2 - 2\gamma(1-t)\langle a, b\rangle + \gamma(1-t)^2\|b\|^2, \\
(t + 2\gamma(1-t))\langle a, b\rangle &\geq \gamma\|a\|^2 + (t(1-t) + \gamma(1-t)^2)\|b\|^2.
\end{aligned} \tag{33}$$

Now note that

$$\langle a, b\rangle = t\langle D_\sigma(x) - D_\sigma(y), x - y\rangle + (1 - t)\|x - y\|^2. \tag{34}$$

Since $D_\sigma$ is $\gamma$-cocoercive with $\gamma > 0$, we know that $D_\sigma$ is $\frac{1}{\gamma}$-Lipschitz. Therefore, by Cauchy-Schwarz inequality,

$$\langle D_\sigma(x) - D_\sigma(y), x - y\rangle \leq \frac{1}{\gamma}\|x - y\|^2. \tag{35}$$

That is,

$$\begin{aligned}
\langle a, b\rangle &\leq \left(\tfrac{t}{\gamma} + 1 - t\right)\|b\|^2, \\
\gamma(1-t)\langle a, b\rangle &\leq t(1-t)\|b\|^2 + \gamma(1-t)^2\|b\|^2.
\end{aligned} \tag{36}$$

By substituting (36) into (33), we have

$$\begin{aligned} (t + \gamma(1-t))\langle a, b\rangle &\geq \gamma\|a\|^2, \\ \langle a, b\rangle &\geq \frac{\gamma}{t + \gamma(1-t)}\|a\|^2, \end{aligned} \tag{37}$$

which means that $D_\sigma^t$ is $\frac{\gamma}{t+\gamma(1-t)}$-cocoercive. Since $D_\sigma$ is conservative, $D_\sigma^t$ is also conservative. By Theorem A.5, we know that there exists a $r$-weakly convex function $F : V \to \bar{\mathbb{R}}$, where

$$r = r(t) = \beta\left(1 - \frac{\gamma}{t + \gamma(1-t)}\right) = \beta\frac{t - \gamma t}{t + \gamma - \gamma t},$$

such that $D_\sigma^t \in \mathrm{Prox}_{\frac{F}{\beta}}$.

Now we prove that $\partial F$ is $L$-Lipschitz. We consider two cases seperately.
**Case 1:** $t \geq \frac{1-2\gamma}{2-2\gamma}$ and $t \in [0, 1)$.

In this case, we need to prove that $\partial F$ is $L(t) = \beta\frac{t}{1-t}$ Lipschitz. Given $\forall x, y \in V$, choose arbitrary $u, v$, such that, $u = D_\sigma^t(x) \in (I + \frac{1}{\beta}\partial F)^{-1}(x), v = D_\sigma^t(y) \in (I + \frac{1}{\beta}\partial F)^{-1}(y)$, then

$$\beta(x - u) \in \partial F(u), \beta(y - v) \in \partial F(v). \tag{38}$$

Note that $a = u - v, b = x - y$. In order to prove $\partial F$ is $L(t)$-Lipschitz, we need to prove

$$\begin{aligned} \|\beta(x-u) - \beta(y-v)\|^2 &\leq L^2(t)\|u - v\|^2 = \beta^2\frac{t^2}{(1-t)^2}\|u - v\|^2, \\ \|\beta(b-a)\|^2 &\leq \beta^2\frac{t^2}{(1-t)^2}\|a\|^2, \\ \|b - a\|^2 &\leq \frac{t^2}{(1-t)^2}\|a\|^2. \end{aligned} \tag{39}$$

Since $a = u - v = D_\sigma^t(x) - D_\sigma^t(y) = t(D_\sigma(x) - D_\sigma(y)) + (1-t)(x-y), b = x - y$, we have

$$a - b = t(D_\sigma(x) - D_\sigma(y)) - t(x - y). \tag{40}$$

Now we only need to prove

$$\begin{aligned} &t^2\|D_\sigma(x) - D_\sigma(y)\|^2 - 2t^2\langle D_\sigma(x) - D_\sigma(y), x - y\rangle + t^2\|x - y\|^2 \\ &\leq \frac{t^2}{(1-t)^2}\left(t^2\|D_\sigma(x) - D_\sigma(y)\|^2 + 2t(1-t)\langle D_\sigma(x) - D_\sigma(y), x - y\rangle + (1-t)^2\|x - y\|^2\right), \end{aligned} \tag{41}$$

which is equivalent to prove

$$\frac{1 - 2t}{2 - 2t}\|D_\sigma(x) - D_\sigma(y)\|^2 \leq \langle D_\sigma(x) - D_\sigma(y), x - y\rangle. \tag{42}$$

Since $t \geq \frac{1-2\gamma}{2-2\gamma}$, we have that

$$\frac{1 - 2t}{2 - 2t} \leq \frac{1 - 2\frac{1-2\gamma}{2-2\gamma}}{2 - 2\frac{1-2\gamma}{2-2\gamma}} = \frac{2 - 2\gamma - 2(1 - 2\gamma)}{2(2 - 2\gamma) - 2(1 - 2\gamma)} = \frac{2\gamma}{2} = \gamma. \tag{43}$$

We already have

$$\gamma\|D_\sigma(x) - D_\sigma(y)\|^2 \leq \langle D_\sigma(x) - D_\sigma(y), x - y\rangle. \tag{44}$$

Thus,

$$\frac{1 - 2t}{2 - 2t}\|D_\sigma(x) - D_\sigma(y)\|^2 \leq \gamma\|D_\sigma(x) - D_\sigma(y)\|^2 \leq \langle D_\sigma(x) - D_\sigma(y), x - y\rangle. \tag{45}$$

**Case 2:** $t \leq \frac{1-2\gamma}{2-2\gamma}$ and $t \in [0, 1)$.

In this case, we need to prove that $\partial F$ is $L(t) = r(t) = \beta \frac{t-t\gamma}{t+\gamma-t\gamma}$ Lipschitz. Similarly in (39), we need to prove

$$
\begin{aligned}
\|\beta(x-u) - \beta(y-v)\|^2 &\le L^2(t)\|u-v\|^2 = \beta^2 \frac{t^2(1-\gamma)^2}{(t+\gamma-t\gamma)^2}\|u-v\|^2, \\
\|b-a\|^2 &\le \frac{t^2(1-\gamma)^2}{(t+\gamma-t\gamma)^2}\|a\|^2.
\end{aligned}
\tag{46}
$$

Since $a = u - v = D_\sigma^t(x) - D_\sigma^t(y) = t(D_\sigma(x) - D_\sigma(y)) + (1-t)(x-y)$, $b = x - y$, we have
$$
a - b = t(D_\sigma(x) - D_\sigma(y)) - t(x-y).
\tag{47}
$$

Now we only need to prove
$$
\begin{aligned}
&t^2\|D_\sigma(x) - D_\sigma(y)\|^2 - 2t^2\langle D_\sigma(x) - D_\sigma(y), x-y\rangle + t^2\|x-y\|^2 \\
\le\ & \frac{t^2(1-\gamma)^2}{(t+\gamma-t\gamma)^2}\left(t^2\|D_\sigma(x) - D_\sigma(y)\|^2 + 2t(1-t)\langle D_\sigma(x) - D_\sigma(y), x-y\rangle + (1-t)^2\|x-y\|^2\right).
\end{aligned}
\tag{48}
$$

When $t = 0$, it holds naturally. When $t \in (0,1)$, and $t \le \frac{1-2\gamma}{2-2\gamma}$, it is equivalent to prove
$$
\begin{aligned}
&\|D_\sigma(x) - D_\sigma(y)\|^2 - 2\langle D_\sigma(x) - D_\sigma(y), x-y\rangle + \|x-y\|^2 \le \\
&\frac{(1-\gamma)^2}{(t+\gamma-t\gamma)^2}\left(t^2\|D_\sigma(x) - D_\sigma(y)\|^2 + 2t(1-t)\langle D_\sigma(x) - D_\sigma(y), x-y\rangle + (1-t)^2\|x-y\|^2\right),
\end{aligned}
\tag{49}
$$

which is equivalent to prove
$$
\begin{aligned}
&(t+\gamma-t\gamma)^2\left[\|D_\sigma(x) - D_\sigma(y)\|^2 - 2\langle D_\sigma(x) - D_\sigma(y), x-y\rangle + \|x-y\|^2\right] \\
\le\ & (1-\gamma)^2\left[t^2\|D_\sigma(x) - D_\sigma(y)\|^2 + 2t(1-t)\langle D_\sigma(x) - D_\sigma(y), x-y\rangle + (1-t)^2\|x-y\|^2\right],
\end{aligned}
\tag{50}
$$

that is to prove
$$
\begin{aligned}
&\left[(t+\gamma-t\gamma)^2 - t^2(1-\gamma)^2\right]\|D_\sigma(x) - D_\sigma(y)\|^2 \\
&- \left[2(t+\gamma-t\gamma)^2 + 2t(1-t)(1-\gamma)^2\right]\langle D_\sigma(x) - D_\sigma(y), x-y\rangle \\
\le\ & \left[(1-t)^2(1-\gamma)^2 - (t+\gamma-t\gamma)^2\right]\|x-y\|^2.
\end{aligned}
\tag{51}
$$

Now we check each coefficient, and estimate each term carefully. For the coefficient of $\|D_\sigma(x) - D_\sigma(y)\|^2$, we have
$$
(t+\gamma-t\gamma)^2 - t^2(1-\gamma)^2 = 2\gamma(1-\gamma)t + \gamma^2 > 0
\tag{52}
$$
For the coefficient of $\langle D_\sigma(x) - D_\sigma(y), x-y\rangle$, it is obviously non-positive. Besides, we have that
$$
\begin{aligned}
&-\left[2(t+\gamma-t\gamma)^2 + 2t(1-t)(1-\gamma)^2\right] \\
=\ & -2\left[(1-\gamma)^2 t^2 + 2\gamma(1-\gamma)t + \gamma^2 + (t-t^2)(1-\gamma)^2\right] \\
=\ & -(2-2\gamma^2)t - 2\gamma^2.
\end{aligned}
\tag{53}
$$
Since $D_\sigma$ is $\gamma$-cocoercive, we have that
$$
\left[-(2-2\gamma^2)t - 2\gamma^2\right]\langle D_\sigma(x) - D_\sigma(y), x-y\rangle \le \left[-\gamma(2-2\gamma^2)t - 2\gamma^3\right]\|D_\sigma(x) - D_\sigma(y)\|^2. \tag{54}
$$
For the coefficient of $\|x-y\|^2$, note that when $\gamma \in (0,1)$, $t \in [0,1)$, and $t \le \frac{1-2\gamma}{2-2\gamma}$, we have
$$
(1-t)^2(1-\gamma)^2 - (t+\gamma-t\gamma)^2 = (2\gamma-2)t - 2\gamma + 1 \ge (2\gamma-2)\frac{1-2\gamma}{2-2\gamma} - 2\gamma + 1 \ge 0. \tag{55}
$$

Combining (52)-(55), to prove (51), we only need to prove that
$$
\begin{aligned}
\left[2\gamma(1-\gamma)t + \gamma^2 - \gamma(2-2\gamma^2)t - 2\gamma^3\right]\|D_\sigma(x) - D_\sigma(y)\|^2 &\le \left[(2\gamma-2)t - 2\gamma + 1\right]\|x-y\|^2 \\
\Longleftrightarrow\ \left[(2\gamma^3 - 2\gamma^2)t + \gamma^2 - 2\gamma^3\right]\|D_\sigma(x) - D_\sigma(y)\|^2 &\le \left[(2\gamma-2)t - 2\gamma + 1\right]\|x-y\|^2.
\end{aligned}
\tag{56}
$$
Since $D_\sigma$ is $\frac{1}{\gamma}$-Lipschitz, we only need to prove that
$$
\begin{aligned}
\frac{1}{\gamma^2}\left[(2\gamma^3 - 2\gamma^2)t + \gamma^2 - 2\gamma^3\right] &\le (2\gamma-2)t - 2\gamma + 1 \\
\Longleftrightarrow\qquad (2\gamma-2)t + 1 - 2\gamma &\le (2\gamma-2)t - 2\gamma + 1.
\end{aligned}
\tag{57}
$$
This completes the proof. $\qquad\square$

Please note that any Lipschitz operator must be single-valued. Thus, there is actually only one element in $\partial F(x)$, given any $x \in V$.

## A.5 Calculation for $\|(2\gamma\,\mathrm{J}-\mathrm{I})\|_*$ and $\|\,\mathrm{J}-\mathrm{J}^\mathrm{T}\,\|_*$

We mainly use the power iterative method [21] to calculate the spectral norm of a given operator A.

---

**Algorithm 1** Power iterative method

---
Given $q^0$ with $\|q^0\|=1$, A, $N$
**for** $n = 1 : N$ **do**
  $z^n = \mathrm{A}\,q^{n-1}$
  $q^n = \frac{z^n}{\|z^n\|}$
**end for**
Return $\lambda^N = (q^N)^\mathrm{T}\,\mathrm{A}\,q^N$

---

By Algorithm 1, we can compute the spectral norm of A. Note that in Algorithm 1, we need to calculate the matrix-vector product $\mathrm{A}\,q$, given any $q \in V$.

Given a denoiser $\mathrm{D}_\sigma$, $\forall x \in V$, $\mathrm{J}(x) = \nabla\,\mathrm{D}_\sigma(x)$ is the Jacobian matrix at the point $x$. To specify, let $x = [x_1, x_2, \ldots, x_n]^\mathrm{T} \in V = \mathbb{R}^n$, and $y = [y_1, y_2, \ldots, y_n]^\mathrm{T} = \mathrm{D}_\sigma(x)$ be the output after $\mathrm{D}_\sigma$. Then the Jacobian matrix $\mathrm{J}(x^*)$ at a given point $x^*$ is:

$$\mathrm{J}(x^*) = \begin{bmatrix} \dfrac{\partial y_1}{\partial x_1} & \dfrac{\partial y_1}{\partial x_2} & \cdots & \dfrac{\partial y_1}{\partial x_n} \\ \dfrac{\partial y_2}{\partial x_1} & \dfrac{\partial y_2}{\partial x_2} & \cdots & \dfrac{\partial y_2}{\partial x_n} \\ \vdots & \vdots & \ddots & \vdots \\ \dfrac{\partial y_n}{\partial x_1} & \dfrac{\partial y_n}{\partial x_2} & \cdots & \dfrac{\partial y_n}{\partial x_n} \end{bmatrix}\Bigg|_{x=x^*} . \tag{58}$$

We start with $a = [a_1, a_2, \ldots, a_n]^\mathrm{T} = \mathrm{J}^\mathrm{T}(x)v$, where $v = [v_1, v_2, \ldots, v_n]^\mathrm{T}$ is any vector. By basic linear algebra, we know that

$$a_i = \sum_{j=1}^n \frac{\partial y_j}{\partial x_i} v_j, \; \forall i = 1, 2, \ldots, n, \tag{59}$$

which implies

$$a = \mathrm{J}^\mathrm{T}(x)v = \frac{\partial\langle y, v\rangle}{\partial x}. \tag{60}$$

The calculation of $b = \mathrm{J}(x)v$ is a bit more complex. We will use the so-called double back-propagation technique: since $b = \mathrm{J}(x)v$, we have

$$b_k = \sum_{i=1}^n \frac{\partial y_k}{\partial x_i} v_i, \; \forall k = 1, 2, \ldots, n. \tag{61}$$

Now consider $w = [w_1, w_2, \ldots, w_n]^\mathrm{T}$. Let $t = \mathrm{J}^\mathrm{T}(x)w$. Thus $t_i = \sum_{j=1}^n \frac{\partial y_j}{\partial x_i} w_j$.

Let $z = \dfrac{\partial\langle t, v\rangle}{\partial w}$, then

$$z = \frac{\partial\langle t, v\rangle}{\partial w} = \frac{\partial}{\partial w}\left(\sum_{i=1}^n v_i \left(\sum_{j=1}^n \frac{\partial y_j}{\partial x_i} w_j\right)\right). \tag{62}$$

The $k$th element of $z$ is:

$$z_k = \frac{\partial}{\partial w_k}\left(\sum_{i=1}^n v_i \left(\sum_{j=1}^n \frac{\partial y_j}{\partial x_i} w_j\right)\right) = \sum_{i=1}^n v_i \frac{\partial y_k}{\partial x_i}, \; \forall k = 1, 2, \ldots, n. \tag{63}$$

That is,

$$z = \mathrm{J}(x)v = \frac{\partial \langle t, v \rangle}{\partial w} = \frac{\partial \langle \frac{\partial \langle y, w \rangle}{\partial x}, v \rangle}{\partial w}, \ \forall w \in \mathbb{R}^n. \tag{64}$$

The AutoGrad toolbox in Pytorch [50] allows the calculation for $\frac{\partial \langle \cdot, \cdot \rangle}{\partial \cdot}$. The pseudo-codes in Pytorch can be:

```
# Calculate J(x)v
def _jacobian_vec(y, x, v):
    w = torch.ones_like(x, require_grad=True)
    t = torch.autograd.grad(y, x, w, create_graph=True)[0]
    return torch.autograd.grad(t, w, v, create_graph=True)[0]

# Calculate J^T(x)v
def _jacobian_transpose_vec(y, x, v):
    return torch.autograd.grad(y, x, v, create_graph=True)[0]
```

By (60) and (64), one can calculate $\|(2\gamma\,\mathrm{J} - \mathrm{I})\|_*$ and $\|\,\mathrm{J} - \mathrm{J}^{\mathrm{T}}\,\|_*$ by Algorithm 1.

## A.6 Proof of Theorem 4.1

We will make use of the Lyapunov function $L_\beta$ for (19) according to [40, 68, 29]:

$$L_\beta(u, v, b) = F(v) + G(u; f) + \beta \langle b, u - v \rangle + \frac{\beta}{2}\|u - v\|^2. \tag{65}$$

We will first prove in part 1 that an important value for $L_\beta(u, v, b)$ is positive whenever $t \in (0, t_0)$, where $t_0$ is the positive root of the characteristic equation in (20). Then, we will prove in part 2 that $L_\beta$ is non-increasing with the iteration number $k$. Finally, we will prove in part 3 that CoCo-ADMM iteration in (19) converges globally to a stationary point of (15).

*Proof.* Let $h(t) = (2 - 2\gamma)t^3 + \gamma t^2 + 2\gamma t - \gamma$, where $\gamma \in (0, 1)$. Note that $h$ is obviously smooth, and $h(0) = -\gamma < 0$, $h(\infty) = \infty$. Also note that when $t > 0$,

$$h'(t) = (2 - 2\gamma)t^2 + 2\gamma t + 2\gamma > 0. \tag{66}$$

Therefore, there exists a unique $t_0 > 0$, such that $h(t_0) = 0$, $h(t) > 0$ if $t > t_0$, and $h(t) < 0$ if $t \in [0, t_0)$.

**Part 1:**

We consider a characteristic value for $L_\beta(u, v, b)$ in (65): $\frac{\beta}{2} - \frac{r}{2} - \frac{L^2}{\beta}$.

By Theorem 2.2, if $t \in [0, 1)$ and $t \geq \frac{1 - 2\gamma}{2 - 2\gamma}$, we have that $r = \beta \frac{t - \gamma t}{t + \gamma - t\gamma}$ and $L = \beta \frac{t}{1 - t}$. Thus we have

$$
\begin{aligned}
\frac{\beta}{2} - \frac{r}{2} - \frac{L^2}{\beta} &= \frac{\beta}{2} - \frac{\beta(t - \gamma t)}{2(t + \gamma - t\gamma)} - \frac{\beta t^2}{(1 - t)^2} \\
&= \frac{\beta}{2}\left(1 - \frac{t - \gamma t}{t + \gamma - t\gamma} - \frac{2t^2}{(1 - t)^2}\right) \\
&= \frac{\beta}{2(t + \gamma - t\gamma)(1 - t)^2}\left(\gamma(1 - t)^2 - 2t^2(t + \gamma - t\gamma)\right) \\
&= -\frac{\beta}{2(t + \gamma - t\gamma)(1 - t)^2}\left((2 - 2\gamma)t^3 + \gamma t^2 + 2\gamma t - \gamma\right).
\end{aligned} \tag{67}
$$

When $0 < t < t_0$, where $t_0$ is the positive root of the characteristic equation in (20), $\frac{\beta}{2} - \frac{r}{2} - \frac{L^2}{\beta} > 0$ holds.

If $t \in [0, 1)$ and $t \leq \frac{1-2\gamma}{2-2\gamma}$, we have that $L = r = \beta \frac{t-\gamma t}{t+\gamma-t\gamma}$. Note that in this case, $L = r \leq \beta \frac{t}{1-t}$. Thus,

$$
\begin{aligned}
& \frac{\beta}{2} - \frac{r}{2} - \frac{L^2}{\beta} \\
\geq \ & \frac{\beta}{2} - \frac{\beta(t - \gamma t)}{2(t + \gamma - t\gamma)} - \frac{\beta t^2}{(1-t)^2} \\
= \ & -\frac{\beta}{2(t + \gamma - t\gamma)(1-t)^2} \left( (2 - 2\gamma)t^3 + \gamma t^2 + 2\gamma t - \gamma \right).
\end{aligned}
\tag{68}
$$

Therefore, we also have that when $0 < t < t_0$, $\frac{\beta}{2} - \frac{r}{2} - \frac{L^2}{\beta} > 0$ holds.

**Part 2:**

Now we prove that $L_\beta(u^k, v^k, b^k)$ is non-increasing. Before that, we show two important formulas. For $v^{k+1}$, by the first-order optimal condition, we know that

$$
\beta b^{k+1} = -\beta(v^{k+1} - u^{k+1} - b^k) \in \partial F(v^{k+1}).
\tag{69}
$$

Similarly, for $u^{k+1}$, we have

$$
-\beta(u^{k+1} - v^k + b^k) \in \partial G(u^{k+1}; f).
\tag{70}
$$

In order to prove that $L_\beta(u^k, v^k, b^k)$ is non-increasing, we decompose $L_\beta(u^k, v^k, b^k) - L_\beta(u^{k+1}, v^{k+1}, b^{k+1})$ into two parts:

$$
\begin{aligned}
& L_\beta(u^k, v^k, b^k) - L_\beta(u^{k+1}, v^{k+1}, b^{k+1}) \\
= \ & L_\beta(u^k, v^k, b^k) - L_\beta(u^{k+1}, v^k, b^k) + L_\beta(u^{k+1}, v^k, b^k) - L_\beta(u^{k+1}, v^{k+1}, b^{k+1}),
\end{aligned}
\tag{71}
$$

and estimate them separately as follows. By the convexity of $G$ and the iteration form in (19), we have

$$
\begin{aligned}
& L_\beta(u^k, v^k, b^k) - L_\beta(u^{k+1}, v^k, b^k) \\
= \ & G(u^k; f) - G(u^{k+1}; f) + \beta\langle b^k, u^k - u^{k+1}\rangle + \frac{\beta}{2}\|u^k - v^k\|^2 - \frac{\beta}{2}\|u^{k+1} - v^k\|^2 \\
= \ & G(u^k; f) - G(u^{k+1}; f) - \langle -\beta(u^{k+1} - v^k + b^k), u^k - u^{k+1}\rangle + \beta\langle v^k - u^{k+1}, u^k - u^{k+1}\rangle \\
& + \frac{\beta}{2}\|u^k - v^k\|^2 - \frac{\beta}{2}\|u^{k+1} - v^k\|^2 \\
\geq \ & 0 + \beta\langle v^k - u^{k+1}, u^k - u^{k+1}\rangle + \frac{\beta}{2}\langle u^k - u^{k+1}, u^k + u^{k+1} - 2v^k\rangle \\
= \ & \beta\langle u^k - u^{k+1}, v^k - u^{k+1} - v^k + \frac{u^k + u^{k+1}}{2}\rangle \\
= \ & \frac{\beta}{2}\|u^k - u^{k+1}\|^2.
\end{aligned}
\tag{72}
$$

By the $r$-weakly convexity of $F$, $\forall x, y \in V$, $f_y \in \partial F(y)$, we have:

$$
F(x) - F(y) \geq \langle f_y, x - y\rangle - \frac{r}{2}\|x - y\|^2.
\tag{73}
$$

By the $L$-Lipschitz property of $\partial F$, $\forall x, y \in V$, $f_y \in \partial F(y)$, we have:

$$
F(x) - F(y) \leq \langle f_y, x - y\rangle + \frac{L}{2}\|x - y\|^2.
\tag{74}
$$

Combining (73) and (74), we can obtain that

$$
\begin{aligned}
& L_\beta(u^{k+1}, v^k, b^k) - L_\beta(u^{k+1}, v^{k+1}, b^{k+1}) \\
=\ & F(v^k) - F(v^{k+1}) + \beta\langle b^k, u^{k+1} - v^k\rangle - \beta\langle b^{k+1}, u^{k+1} - v^{k+1}\rangle \\
& + \frac{\beta}{2}\|u^{k+1} - v^k\|^2 - \frac{\beta}{2}\|u^{k+1} - v^{k+1}\|^2 \\
=\ & F(v^k) - F(v^{k+1}) + \beta\langle b^k, u^{k+1} - v^k\rangle - \beta\langle b^k, u^{k+1} - v^{k+1}\rangle - \beta\langle u^{k+1} - v^{k+1}, u^{k+1} - v^{k+1}\rangle \\
& + \frac{\beta}{2}\|v^k - v^{k+1}\|^2 + \beta\langle u^{k+1} - v^{k+1}, v^{k+1} - v^k\rangle \\
=\ & F(v^k) - F(v^{k+1}) + \beta\langle b^k, v^{k+1} - v^k\rangle - \beta\|u^{k+1} - v^{k+1}\|^2 \\
& + \frac{\beta}{2}\|v^k - v^{k+1}\|^2 + \beta\langle u^{k+1} - v^{k+1}, v^{k+1} - v^k\rangle \\
=\ & F(v^k) - F(v^{k+1}) - \langle \beta b^k, v^{k+1} - v^k\rangle - \beta\|u^{k+1} - v^{k+1}\|^2 + \frac{\beta}{2}\|v^k - v^{k+1}\|^2 \\
& + \beta\langle u^{k+1} - v^{k+1}, v^{k+1} - v^k\rangle \\
\geq\ & -\frac{r}{2}\|v^k - v^{k+1}\|^2 - \beta\|u^{k+1} - v^{k+1}\|^2 + \frac{\beta}{2}\|v^k - v^{k+1}\|^2 \\
=\ & \left(\frac{\beta}{2} - \frac{r}{2}\right)\|v^k - v^{k+1}\|^2 - \beta\|b^k - b^{k+1}\|^2 \\
\geq\ & \left(\frac{\beta}{2} - \frac{r}{2}\right)\|v^k - v^{k+1}\|^2 - \frac{L^2}{\beta}\|v^k - v^{k+1}\|^2 \\
=\ & \left(\frac{\beta}{2} - \frac{r}{2} - \frac{L^2}{\beta}\right)\|v^k - v^{k+1}\|^2.
\end{aligned}
\tag{75}
$$

Note that the second '=' comes from the cosine rule, the first '≥' follows from the $r$-weakly convexity of $F$ as in (73), and the second '≥' results from the $L$-Lipschitz of $\partial F$ as in (74).

Combining (72) and (75), we get

$$
L_\beta(u^k, v^k, b^k) - L_\beta(u^{k+1}, v^{k+1}, b^{k+1}) \geq \frac{\beta}{2}\|u^k - u^{k+1}\|^2 + \left(\frac{\beta}{2} - \frac{r}{2} - \frac{L^2}{\beta}\right)\|v^k - v^{k+1}\|^2 \geq 0,
\tag{76}
$$

that is, $L_\beta(u^k, v^k, b^k)$ is non-increasing.

Now we prove that $\{(u^k, v^k, b^k)\}$ is bounded. Note that $F$ and $G$ are coercive on $V$. As a result,

$$
F(u^k) + G(u^k; f) > +\infty.
\tag{77}
$$

Along with $\beta b^k \in \partial F(v^k)$ and the property that $\partial F$ is $L$-Lipschitz, we arrive at

$$
\begin{aligned}
L_\beta(u^k, v^k, b^k) &= F(v^k) + G(u^k; f) + \beta\langle b^k, u^k - v^k\rangle + \frac{\beta}{2}\|u^k - v^k\|^2 \\
&\geq F(u^k) + G(u^k; f) - \frac{L}{2}\|u^k - v^k\|^2 + \frac{\beta}{2}\|u^k - v^k\|^2.
\end{aligned}
\tag{78}
$$

Note that $L = \frac{\beta t}{1-t}$, and that $t < 0.5$. Thus, $L < \beta$. Therefore,

$$
\begin{aligned}
& F(u^k) + G(u^k; f) - \frac{L}{2}\|u^k - v^k\|^2 + \frac{\beta}{2}\|u^k - v^k\|^2 \\
=\ & F(u^k) + G(u^k; f) + \frac{1}{2}(\beta - L)\|u^k - v^k\|^2 \geq -\infty.
\end{aligned}
\tag{79}
$$

Since $F(u) + G(u; f)$ is coercive on $V$, $u^k, v^k, b^k$ are bounded.
**Part 3:**

Define $q^{k+1}$ as follows:

$$
q^{k+1} = [\beta(b^{k+1} - b^k + v^k - v^{k+1}), \beta(v^{k+1} - u^{k+1}), \beta(b^{k+1} - b^k)].
\tag{80}
$$

Define $\partial L_\beta(u, v, b) = [\partial_u L_\beta, \partial_v L_\beta, \partial_b L_\beta]$. By the formulas (69)-(70), we know that

$$
q^{k+1} \in \partial L_\beta(u^{k+1}, v^{k+1}, b^{k+1}).
\tag{81}
$$

Note that
$$\|\beta(b^{k+1}-b^k+v^k-v^{k+1})\| \le \beta\|b^k-b^{k+1}\|+\beta\|v^k-v^{k+1}\| \le L\|v^k-v^{k+1}\|+\beta\|v^k-v^{k+1}\|, \quad (82)$$
and that
$$\|\beta(v^{k+1}-u^{k+1})\| = \beta\|b^k-b^{k+1}\| \le L\|v^k-v^{k+1}\|, \quad (83)$$
we arrive at
$$\|q^{k+1}\| \le C\|v^k-v^{k+1}\|, \quad (84)$$
where
$$C = 3L+\beta. \quad (85)$$

Now we can finally prove Theorem 4.1. By Part 2, $\{(u^k,v^k,b^k)\}$ is bounded. So there is a sub-sequence $\{(u^{n_k},v^{n_k},b^{n_k})\}$ such that $(u^{n_k},v^{n_k},b^{n_k}) \to (u^*,v^*,b^*)$, when $n \to +\infty$. Since $L_\beta(u^k,v^k,b^k)$ is lower bounded and non-increasing, we have that $\|u^k-u^{k+1}\|, \|v^k-v^{k+1}\| \to 0$ as $k \to +\infty$. Besides, since $q^k \in \partial L_\beta(u^k,v^k,b^k)$ and $\|q^k\| \le C\|v^k-v^{k+1}\|$, we know that $\|q^k\| \to 0, \|q^{n_k}\| \to 0$. Thus, $0 \in \partial L_\beta(u^*,v^*,b^*)$, and $(u^*,v^*,b^*)$ is a stationary point of $L_\beta$.

Since $F,G$ is KL, we conclude that $L_\beta$ is also KL. Then, by the proof of Theorem 2.9 in [2], $\{(u^{n_k},v^{n_k},b^{n_k})\}$ converges globally to $(u^*,v^*,b^*)$.

Since $(u^*,v^*,b^*)$ is a stationary point of $L_\beta$, and as a result, $q^* = 0$, that is
$$q^* = [0,\beta(v^*-u^*),0] = 0. \quad (86)$$
Therefore, $u^* = v^*$. By CoCo-ADMM iteration in (19), we know that
$$\begin{aligned} u^* &= \operatorname{Prox}_{\frac{G}{\beta}}(u^*-b^*), \\ u^* &= \mathrm{D}_\sigma^t(u^*+b^*) \in \operatorname{Prox}_{\frac{G}{\beta}}(u^*+b^*). \end{aligned} \quad (87)$$

Equivalently,
$$-\beta b^* \in \partial G(u^*;f), \beta b^* \in \partial F(u^*), \quad (88)$$

$$0 \in \partial F(u^*) + \partial G(u^*;f). \quad (89)$$
Therefore, $u^*$ is a stationary point of (15). □

## A.7 Proof of Theorem 4.4

The convergence of PGD with weakly convex prior function has been extensively studied. For details, one can refer to [2, 27]. For example, [27] prove that

**Theorem A.6** (Theorem 1 by [27]). *Let $F,H$ be proper, closed, bounded from below, and $H$ is differentiable with $L_H$-Lipschitz gradient, and $F$ is $r$-weakly convex. Then for $\tau < \max\{\frac{2}{L_H+r},\frac{1}{L_H}\}$, the iterates*
$$u^{k+1} \in \operatorname{Prox}_{\tau F}\circ(\mathrm{I}-\tau\nabla H)(u^k) \quad (90)$$
*verify*

- *$(F(u^k)+H(u^k))$ is non-increasing and converges.*

- *All cluster points of the sequence $u^k$ are stationary points of $F+H$.*

- *If the sequence $u^k$ is bounded and if $F+H$ verifies the KL property at the cluster points of $u^k$, then $u^k$ converges to a stationary point of $F+H$.*

By Theorem A.6, we can prove Theorem 4.4.

*Proof.* Let $F$ be the proper, closed, weakly convex prior function. Let $H = {}^1G$ be the Moreau envelope of $G$, that is
$$H(u) = {}^1G(u) := \min_v G(v) + \frac{1}{2}\|v-u\|^2. \quad (91)$$
Then ${}^1G$ is diffentiable:
$$\nabla {}^1G = \mathrm{I} - \operatorname{Prox}_G. \quad (92)$$
Since $G$ is proper, closed, and convex, its proximal operator is firmly non-expansive. Therefore, $\nabla {}^1G$ is also firmly non-expansive, thus 1-Lipschitz. Since $F$ and $G$ is coercive and KL, $F+{}^1G$ is also coercive and KL. Therefore, $u^k$ is bounded. By Theorem A.6, if $\frac{1}{\beta} < \max\{\frac{2}{1+r},1\}$, CoCo-PEGD converges to a stationary point of $F+{}^1G$. □

## A.8 Proximal operator on the fidelity for each task

When $K \neq I$, in general, there is no closed-form solution for $\text{Prox}_{\frac{G}{\beta}}$. Given $f \in V$, we say $\hat{x} = \text{Prox}_{\frac{G}{\beta}}(z)$, if

$$\hat{x} = \arg\min_{x} \lambda\langle\mathbf{1}, Kx - f\log Kx\rangle + \frac{\beta}{2}\|x - z\|^2. \tag{93}$$

(93) is solved by ADMM with $T$ iterations and $\rho > 0$ as follows:

$$\begin{aligned}
x^{i+1} &= (\beta + \rho K^\top K)^{-1}(\beta z + \rho K^\top(y^i - w^i)), \\
y^{i+1} &= \arg\min_{y} \lambda\langle\mathbf{1}, y - f\log y\rangle + \frac{\rho}{2}\|Kx^{i+1} - y + w^i\|^2, \\
w^{i+1} &= w^i + Kx^{i+1} - y^{i+1}.
\end{aligned} \tag{94}$$

When $K$ is a blur kernel, $x$-subproblem can be solved by the fast Fourier transformation as in [49]. When $K = R$ is the Radon transform, one can calculate $(\beta + \rho K^T K)^{-1}$ by the conjugate gradient method. There is a closed form solution for the $y$-subproblem by [34]:

$$y^{i+1} = \frac{z^i + \sqrt{(z^i)^2 + 4\rho\lambda f}}{2\rho}, \tag{95}$$

where

$$z^i = \rho(Kx^{i+1} + \omega^i) - \lambda\mathbf{1}. \tag{96}$$

We initialize $x^0 = y^0 = z, w^0 = 0$, and output $\hat{x} = x^T$. We set $T = 10$ to ensure the convergence of the $x$-subproblem.

## A.9 Detailed Deconvolution results on each kernel

See Tables 3-4.

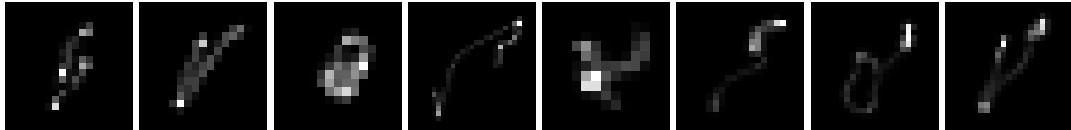

Figure 4: Eight blur kernels from [39].

Table 3: Average Deconvolution PSNR and SSIM performance by different methods on CBSD68 dataset with Poisson noise with peak value $p = 100$.

| p=100 | kernel1 | kernel2 | kernel3 | kernel4 | kernel5 | kernel6 | kernel7 | kernel8 | Average |
|---|---|---|---|---|---|---|---|---|---|
| DPIR | 26.24 | 25.97 | 26.66 | 25.65 | 27.61 | 27.42 | 26.52 | 26.00 | 26.51 |
|  | **0.7321** | **0.7209** | **0.7439** | 0.7016 | **0.7870** | **0.7780** | **0.7462** | **0.7257** | **0.7419** |
| RMMO-DRS | 25.60 | 25.39 | 26.07 | 25.19 | 26.92 | 26.59 | 26.06 | 25.67 | 25.94 |
|  | 0.6895 | 0.6804 | 0.7057 | 0.6665 | 0.7426 | 0.7296 | 0.7089 | 0.6921 | 0.7019 |
| Prox-DRS | 25.32 | 25.03 | 25.82 | 24.80 | 26.75 | 26.39 | 25.94 | 25.40 | 25.68 |
|  | 0.6546 | 0.6488 | 0.6842 | 0.6336 | 0.7256 | 0.7050 | 0.6924 | 0.6668 | 0.6764 |
| DPS | 23.40 | 23.23 | 24.19 | 22.86 | 24.74 | 24.06 | 23.60 | 23.10 | 23.65 |
|  | 0.5941 | 0.5882 | 0.6303 | 0.5699 | 0.6550 | 0.6265 | 0.6039 | 0.5815 | 0.6062 |
| DiffPIR | 24.46 | 24.45 | 25.03 | 24.17 | 25.66 | 25.27 | 24.93 | 24.56 | 24.82 |
|  | 0.6250 | 0.6251 | 0.6510 | 0.6085 | 0.6838 | 0.6664 | 0.6509 | 0.6325 | 0.6429 |
| SNORE | 25.75 | 25.99 | 26.56 | 25.70 | 27.13 | 26.85 | 26.55 | 26.09 | 26.33 |
|  | 0.6985 | 0.6993 | 0.7202 | 0.6856 | 0.7485 | 0.7405 | 0.7249 | 0.7085 | 0.7158 |
| PnPI-HQS | 26.25 | 25.95 | 26.32 | 25.70 | 27.32 | 27.15 | 26.55 | 26.15 | 26.42 |
|  | 0.7024 | 0.6930 | 0.7124 | 0.6797 | 0.7570 | 0.7466 | 0.7269 | 0.7084 | 0.7158 |
| B-RED | 23.89 | 23.55 | 24.09 | 23.33 | 25.04 | 24.61 | 24.33 | 23.88 | 24.09 |
|  | 0.6206 | 0.6188 | 0.6537 | 0.5988 | 0.6824 | 0.6584 | 0.6434 | 0.6200 | 0.6370 |
| CoCo-ADMM | **26.66** | **26.46** | **26.93** | **26.23** | **27.78** | **27.52** | **26.96** | **26.59** | **26.89** |
|  | 0.7270 | 0.7183 | 0.7343 | **0.7075** | 0.7707 | 0.7630 | 0.7404 | 0.7254 | 0.7358 |

Table 4: Average Poisson deblurring PSNR and SSIM performance by different methods on CBSD68 dataset with Poisson noise with peak value $p = 50$.

| p=50 | kernel1 | kernel2 | kernel3 | kernel4 | kernel5 | kernel6 | kernel7 | kernel8 | Average |
|------|---------|---------|---------|---------|---------|---------|---------|---------|---------|
| DPIR | 25.10 | 24.83 | 25.66 | 24.50 | 26.45 | 26.20 | 25.41 | 24.89 | 25.38 |
|      | 0.6773 | 0.6656 | 0.6988 | 0.6480 | 0.7390 | 0.7261 | 0.6981 | 0.6752 | 0.6910 |
| RMMO-DRS | 24.86 | 24.73 | 25.39 | 24.51 | 25.84 | 25.52 | 25.12 | 24.83 | 25.10 |
|      | 0.6439 | 0.6419 | 0.6690 | 0.6292 | 0.6826 | 0.6697 | 0.6549 | 0.6457 | 0.6546 |
| Prox-DRS | 24.89 | 24.69 | 25.45 | 24.43 | 26.14 | 25.81 | 25.36 | 24.92 | 25.21 |
|      | 0.6363 | 0.6319 | 0.6676 | 0.6164 | 0.7024 | 0.6829 | 0.6714 | 0.6484 | 0.6572 |
| DPS | 22.92 | 22.76 | 23.72 | 22.34 | 24.13 | 23.48 | 22.98 | 22.49 | 23.10 |
|      | 0.5724 | 0.5668 | 0.6088 | 0.5475 | 0.6266 | 0.5990 | 0.5758 | 0.5561 | 0.5816 |
| DiffPIR | 23.79 | 23.73 | 24.43 | 23.45 | 24.91 | 24.45 | 24.16 | 23.77 | 24.08 |
|      | 0.5915 | 0.5894 | 0.6220 | 0.5744 | 0.6478 | 0.6263 | 0.6134 | 0.5941 | 0.6074 |
| SNORE | 24.57 | 25.06 | 25.80 | 24.75 | 25.78 | 25.34 | 25.34 | 25.01 | 25.21 |
|      | 0.6522 | 0.6617 | 0.6893 | 0.6490 | 0.6967 | 0.6863 | 0.6749 | 0.6648 | 0.6719 |
| PnPI-HQS | 25.54 | 25.29 | 25.80 | 25.02 | 25.60 | 26.37 | 25.83 | 25.41 | 25.61 |
|      | **0.6972** | 0.6705 | 0.6943 | 0.6563 | 0.7359 | 0.7231 | 0.7039 | 0.6837 | 0.6956 |
| B-RED | 23.49 | 23.41 | 23.86 | 22.99 | 24.69 | 24.49 | 23.86 | 23.46 | 23.78 |
|      | 0.6083 | 0.6059 | 0.6436 | 0.5854 | 0.6692 | 0.6441 | 0.6264 | 0.6027 | 0.6232 |
| CoCo-ADMM | **27.74** | **25.57** | **26.16** | **25.32** | **26.89** | **26.52** | **26.09** | **25.67** | **26.00** |
|      | 0.6906 | **0.6817** | **0.7072** | **0.6701** | **0.7412** | **0.7289** | **0.7098** | **0.6911** | **0.7026** |

## A.10 Single photon imaging

We test the proposed methods in a low-light real-world scenario: single photon imaging task by a time-correlated single-photon avalanche diode (SPAD) camera [62]. In this task, a SPAD array is used to track periodic light pulses in flight, and each detector only recieves about 1 signal photon per pixel on average ($p \approx 1$). By the uncertainty principle [26, 56], since the momentum of the photons is known, the position of the photons cannot be accurately recorded. Therefore, the arrival time of each photon in the SPAD array is a random variable, and can be modelled by a Poisson process. By the filtered histogram method, a reflectivity image[5] is obtained, see Fig. 3 (a).

We compare the visual denoising performance on Fig. 3 (a) by some state-of-the-art Poisson noise removal methods: Photon TV using a TV prior specialized for this problem [62]; BM3D-VST, a BM3D method with variance stabilization transform designed for Poisson noises [3]; DnCNN [74] trained as a Poisson denoiser; VDIR, a variational inference network [63]; VBDNet, a variational bayesian deep network for blind poisson denoising [41]. We also compare three PnP methods, DPIR, RMMO-DRS, and PnPI-HQS, as introduced before.

We show the visual results in Fig. 3. It can be seen in Figs. 3 (b)-(c) that, non-deep methods provide blurred results with unclear edges. End-to-end deep learning methods provide over-smoothed results, and cannot restore the fine details in the flower, see Figs. 3 (d)-(f). Compared with other PnP methods, CoCo methods can recover the fine textures in the flower, as well as the sharp edges in the equation, see Figs. 3 (g)-(k). Since there is no reference image in this real-world scenario, we only report the relative error curves to validate the convergence in Fig. 3 (l).

## A.11 Ablation study on $\gamma$ and $t$ in CoCo-PnP

We report the PSNR values by CoCo-PnP with different $\gamma$ and $t$ in the deblurring task in Table 5. It can be seen that, the proposed methods are sensitive to $\gamma$, because it determines the denoising performance, but not sensitive to $t$.

## A.12 Low-dose CT reconstruction

In order to further illustrate the effectiveness of the proposed methods, we consider the sparse-view low-dose CT reconstruction task. In this task, $K = R$ is the Radon transform, and $f$ in Eq. (1) is the down-sampled data in the Radon field. The Radon field data is corrupted by the Poisson noise.

---

[5]The data is kindly provided by [62] in github.com/photon-efficient-imaging/single-photon-camera.

Table 5: PSNR results by CoCo-PnP with different $\gamma$ and $t$ when deblurring the image '0005' from CBSD68 and kernel 8 from Levin's dataset. The Poisson noise level is 50. We run the algorithm for 500 iterations to ensure the convergence. When $\gamma = 0.25$, $t_0(\gamma) = 1/3$, thus $t < 1/3$. When $\gamma = 0.5$, $t_0(\gamma) \approx 0.3761$, thus $t \le 0.3761$.

| $\gamma = 0.25$ | $t$ | 0.3333 | 0.300 | 0.200 | 0.100 | 0.001 |
|---|---|---|---|---|---|---|
| CoCo-ADMM | PSNR | 25.33 | 25.31 | 25.30 | 25.24 | 25.16 |
| CoCo-PEGD | PSNR | 25.32 | 25.30 | 25.27 | 25.22 | 24.98 |
| $\gamma = 0.50$ | $t$ | 0.3761 | 0.300 | 0.200 | 0.100 | 0.001 |
| CoCo-ADMM | PSNR | 24.88 | 24.87 | 24.79 | 24.64 | 24.47 |
| CoCo-PEGD | PSNR | 24.84 | 24.82 | 24.71 | 24.54 | 24.35 |

For the test set, we select ten typical images from "the NIH-AAPM-Mayo Clinic Low Dose CT Grand Challenge" [45], see Fig. 5.

For the comparison methods, we use: FBP, the conventional filtered back projection method; PWLS-TGV, penalized weighted least-squares (PWLS) with total generalized variation (TGV) prior [48]; PWLS-CSCGR, PWLS with convolutional sparse coding and gradient regularization [5]; UNet, which take the result by FBP as the input [31]; WNet, which uses two UNets to construct the data from both the image and Radon domain [14].

We show the reconstruction results with 60 projection views and Poisson noises with peak value $p = 100, 500$ in Table 2. It can be seen that, compared with two iterative PWLS-based methods, and two end-to-end deep learning methods, the proposed CoCo methods have a significant improvement in PSNR and SSIM values.

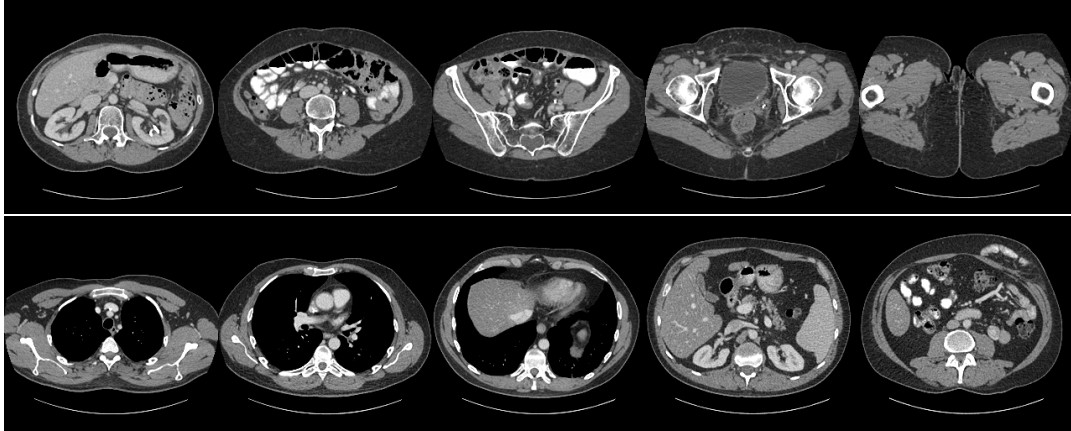

Figure 5: CT test images.

### A.13 Poisson denoising results

We apply CoCo-ADMM and CoCo-PEGD to Poisson denoising problems. In this task, $K = I$ is the identity. The average PSNR and SSIM values on CBSD68 are summarized in Table 6. It can be seen that compared with other convergent methods, the proposed methods achieve the best PSNR and SSIM values. It validates the effectiveness of CoCo methods.

### A.14 Computational cost

In Table 7, we give the average computation time in seconds, as well as the memory cost in MB by different methods when deblurring a $256 \times 256$ image. It can be seen that the proposed methods are the most efficient with least memory cost.

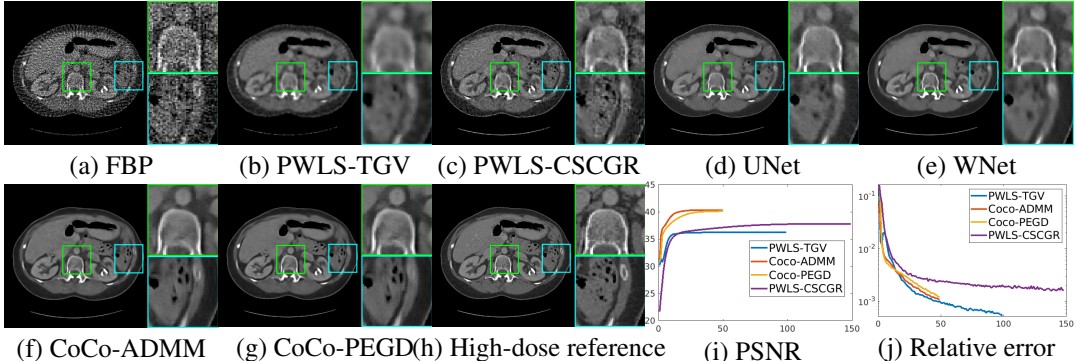

(a) FBP    (b) PWLS-TGV    (c) PWLS-CSCGR    (d) UNet    (e) WNet

(f) CoCo-ADMM    (g) CoCo-PEGD(h) High-dose reference    (i) PSNR    (j) Relative error

Figure 6: Sparse view CT reconstruction results with 60 views and Poisson noises (peak=500). (a) FBP, PSNR=30.91dB. (b) PWLS-TGV, PSNR=36.12dB. (c) PWLS-CSCGR, PSNR=37.80dB. (d) UNet, PSNR=38.54dB. (e) WNet, PSNR=38.74dB. (f) CoCo-ADMM, PSNR=40.46dB. (g) CoCo-PEGD, PSNR=40.28dB. (h) High-dose reference. (i) PSNR curves. (j) Relative error curves.

Table 6: Average denoising PSNR and SSIM performance by different methods on CBSD68 with peak value $p = 30$ and $p = 20$.

|  | $p = 30$ | | $p = 20$ | |
|---|---|---|---|---|
|  | PSNR | SSIM | PSNR | SSIM |
| DPIR | 29.85 | **0.8654** | 28.68 | 0.8300 |
| RMMO-DRS | 27.90 | 0.8008 | 27.74 | 0.7887 |
| Prox-DRS | 28.99 | 0.8087 | 27.94 | 0.7724 |
| SNORE | 29.47 | 0.8474 | 28.34 | 0.8226 |
| PnPI-HQS | 29.90 | 0.8613 | 28.63 | 0.8156 |
| B-RED | 28.80 | 0.8041 | 26.26 | 0.7362 |
| CoCo-ADMM | **29.99** | 0.8604 | 28.76 | 0.8329 |
| CoCo-PEGD | **29.99** | 0.8625 | **28.77** | **0.8332** |

## A.15 Performances under extreme conditions

In Fig. 7, we show an example when deblurring an image under severe noisy condition. It can be seen in Fig. 7 (a) that the image is severely corrupted. We compare results by different methods. It can be seen that CoCo-ADMM and CoCo-PEGD provide sharper edges. Even in this extreme case, the proposed methods are still convergent, see Figs. 7 (m)-(n).

## A.16 Ablation study on the parameters $\alpha_1$ and $\alpha_2$

$$Loss(\theta) = \mathbb{E}\| D_\sigma(x + \xi; \theta) - x\|_1 + \alpha_1 \| J - J^\top \|_* + \alpha_2 \max\{\|2\gamma J - I\|_*, 1 - \epsilon\}. \quad (97)$$

In the loss function (97), $\alpha_1$ controls the penalty strength of the Hamiltonian term to encourage conservativeness, and $\alpha_2$ controls the penalty strength of the spectral term to encourage cocoerciveness.

As shown in Table 8, DRUNet without regularization terms has an expansive residual part. When $\alpha_1$ gets bigger, the mean symmetry error gets lower. When $\alpha_2$ is smaller than 1e-2, the regularized denoiser may not be cocoercive.

|  | B-RED | Prox-DRS | SNORE | DiffPIR | DPS | CoCo-ADMM | CoCo-PEGD |
|---|---|---|---|---|---|---|---|
| Iteration | 300 | 100 | 200 | 100 | 1000 | 50 | 100 |
| Time | 42.5s | 13.6s | 74.1s | 11.8s | 177s | 5.24s | 9.08s |
| Memory | 4946MB | 3982MB | 3624MB | 2998MB | 4694MB | 2304MB | 2304MB |

Table 7: Average computation time in seconds, and memory cost in MB by different methods when deblurring a $256 \times 256$ image.

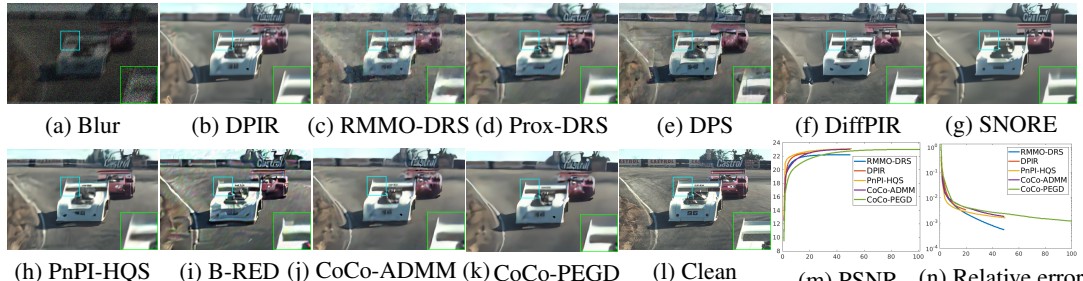

(a) Blur    (b) DPIR    (c) RMMO-DRS   (d) Prox-DRS    (e) DPS    (f) DiffPIR    (g) SNORE

(h) PnPI-HQS    (i) B-RED   (j) CoCo-ADMM   (k) CoCo-PEGD    (l) Clean    (m) PSNR    (n) Relative error

Figure 7: Deconvolution results by different methods on the image '0005' from CBSD68 with kernel 8 and $p = 10$ Poisson noises. (a) Blur image. (b) DPIR, PSNR=22.49dB. (c) RMMO-DRS, PSNR=22.23dB. (d) Prox-DRS, PSNR=22.54dB. (e) DPS, PSNR=20.70dB. (f) Diff-PIR, PSNR=21.66dB. (g) SNORE, PSNR=23.31dB. (h) PnPI-HQS, PSNR=23.14dB. (i) B-RED, PSNR=21.15dB. (j) CoCo-ADMM, PSNR=23.04dB. (k) CoCo-PEGD, PSNR=23.02dB. (l) Clean image. (m) PSNR curves. (n) Relative error curves. $x$-axis denotes the iteration number.

Therefore, we set the penalty parameters large enough to encourage the properties we want. However, when $\alpha$ gets too large, the denoising performance may be sacrificed. We empirically set $\alpha_1 = 1$, $\alpha_2 = $ 1e-2 in the experiments.

Table 8: Mean symmetry error $\|J - J^\top\|_*$ with $N = 1$ and maximal values of the norm $\|2\gamma J - I\|_*$ with $N = 30$ on CBSD68 for various noise levels $\sigma$ and $\gamma = 0.50, 0.25$.

|  | 15 | 25 | 40 | Norms | $\alpha_1$ | $\alpha_2$ |
|---|---|---|---|---|---|---|
| DRUNet | 4.1e+0 | 4.2e+1 | 1.8e+1 | $\|J - J^\top\|_*$ | 0 | 0 |
| 0.50-CoCo-DRUNet | 8.6e-3 | 2.5e-4 | 7.1e-4 | $\|J - J^\top\|_*$ | 1 | 1e-2 |
| 0.50-CoCo-DRUNet | 2.9e-1 | 2.6e-2 | 9.5e-2 | $\|J - J^\top\|_*$ | 1e-2 | 1e-2 |
| 0.25-CoCo-DRUNet | 3.1e-4 | 1.8e-4 | 3.9e-4 | $\|J - J^\top\|_*$ | 1 | 1e-2 |
| 0.25-CoCo-DRUNet | 5.9e-1 | 8.5e-1 | 7.9e-1 | $\|J - J^\top\|_*$ | 1e-2 | 1e-2 |
| DRUNet | 3.285 | 4.343 | 6.283 | $\|J - I\|_*$ | 0 | 0 |
| 0.50-CoCo-DRUNet | 0.994 | 0.992 | 0.972 | $\|J - I\|_*$ | 1 | 1e-2 |
| 0.50-CoCo-DRUNet | 1.011 | 1.244 | 1.500 | $\|J - I\|_*$ | 1 | 1e-3 |
| 0.25-CoCo-DRUNet | 0.986 | 0.969 | 0.982 | $\|0.5\,J - I\|_*$ | 1 | 1e-2 |
| 0.25-CoCo-DRUNet | 0.992 | 1.010 | 1.211 | $\|0.5\,J - I\|_*$ | 1 | 1e-3 |

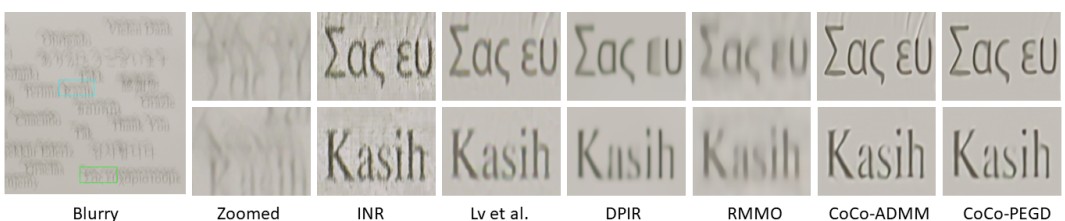

Blurry    Zoomed    INR    Lv et al.    DPIR    RMMO    CoCo-ADMM    CoCo-PEGD

Figure 8: Blind decovolution results by different methods.

## A.17 Blind Deblur Results

In this section, we evaluate the performance of our proposed methods on real-world blind deblurring. We select 18 images from the dataset by Lai et al. [35], which captures challenging real-world blurry scenes. Four state-of-the-art methods are benchmarked: Deblur-INR [76], Lv et al. [42], DPIR [77], and RMMO [66]. Deblur-INR is a self-supervised method that jointly estimates clean images and blur kernels, while Lv et al. [42] applies total variation regularization to refine latent image estimation in a blind deblurring framework. In contrast, DPIR, RMMO, and our proposed CoCo-ADMM/CoCo-PEGD require pre-estimated blur kernels as input. To ensure fairness, all

| | INR | Lv et al. | DPIR | RMMO | CoCo-ADMM | CoCo-PEGD |
|---|---|---|---|---|---|---|
| Average | 0.2990 | 0.3444 | 0.3629 | 0.2664 | 0.5461 | **0.5474** |

Table 9: Average CLIP-IQA results across all deblurred images

kernel-dependent methods utilize blur kernels generated by the blind deblurring framework of Liu et al. [42]. Visual comparisons in Fig. 8 demonstrate that CoCo-ADMM and CoCo-PEGD produce sharper textures and fewer artifacts than baseline methods.

For quantitative evaluation, we employ the CLIP-IQA metric [70], a non-reference image quality assessment tool that leverages the pretrained vision-language representation of CLIP (Contrastive Language–Image Pretraining) to evaluate perceptual quality without requiring pristine reference images. The average CLIP-IQA scores across all deblurred images are summarized in Table 9. Despite relying on the same estimated kernels as DPIR and RMMO, our methods achieve superior performance, highlighting their robustness to kernel estimation errors and enhanced restoration capability.

### A.18 Blind Denoise Results

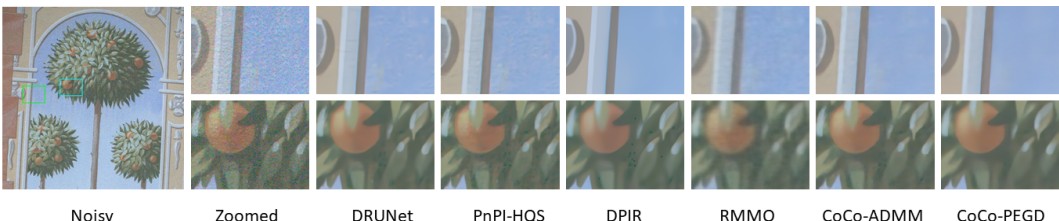

Figure 9: Blind noise removal results by different methods.

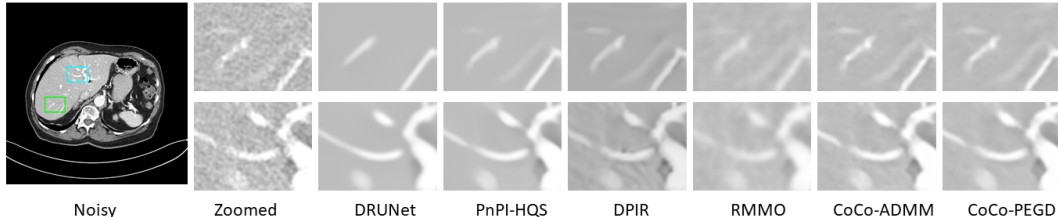

Figure 10: Blind noise suppression results in CT images by different methods.

In this section, we evaluate the performance of our proposed denoising framework on two benchmark datasets that capture real-world noise characteristics. For medical imaging, we employ the Low-Dose CT dataset from Mayo Clinic [46], which provides paired low-dose CT scans with Poisson noise.

For natural images, we utilize the DND dataset [52], comprising 50 real noisy photographs corrupted by a mixture of Poisson and Gaussian noise. Following the standard evaluation protocol, perceptual quality is quantified using the CLIP-IQA metric [70].

To ensure a fair comparison, we benchmark our proposed methods (CoCo-ADMM and CoCo-PEGD) against five representative techniques: DRUNet[77], DPIR [77] , RMMO [66] , and PnPI-HQS [71]. Quantitative results in Table 10 demonstrate that our proposed methods achieve state-of-the-art performance across both imaging domains.

|            | DRUNet | DPIR   | RMMO   | PnPI-HQS | CoCo-ADMM | CoCo-PEGD |
|------------|--------|--------|--------|----------|-----------|-----------|
| Color images | 0.4464 | 0.4509 | 0.4561 | 0.4586   | **0.4596** | 0.4593    |
| CT         | 0.4154 | 0.4258 | 0.4398 | 0.4011   | 0.4707    | **0.4730** |

Table 10: Average CLIP-IQA results for blind image denoising.

