# OpenReview forum: "Learning Cocoercive Conservative Denoisers via Helmholtz Decomposition for Poisson Imaging Inverse Problems"
_NeurIPS.cc/2025/Conference — NeurIPS 2025 poster_

### Official Review · Reviewer_aRzT · 2025-06-27

**Clarity:** 3
**Significance:** 2
**Originality:** 3
**Rating:** 5
**Confidence:** 3

**Summary:**

This paper aims to build better denoising priors for image reconstruction in Poisson inverse problems while maintaining a proximal structure that provides strong theoretical guarantees on convergence. Specifically, relaxes the current strong convergence requirements for denoisers for non-smooth likelihoods (such as those in Poisson inverse problems) in order to be less restrictive, potentially allowing better denoising performance. To this end, the paper introduces a new assumption, called $\gamma$-cocoercivity, which can be made weaker than firm non-expansiveness to allow more freedom for denoiser learning. Further, it requires that the denoiser be a conservative operator in order to avoid wasteful optimization steps. With these assumptions, the authors prove that the denoiser is a proximal operator of an $L$-Lipschitz function. In order to encourage such cocoerciveness and conservativeness properties, they introduce appropriate regularization terms in the denoiser training. They further prove the global convergence of ADMM and proximal gradient descent using this trained denoiser to a stationary point of the Poisson inverse problem. Results on photon-limited inverse problems show the benefits of designing less restricted denoisers while maintaining a guaranteed convergence structure.

**Questions:**

Please see the weaknesses above. Specifically, please clarify:
- the *enforce* v/s *encourage* difference between the theoretical analysis and the training strategy
- if there is a way to quantify how non-conservative a denoiser can be to still be a “good enough” proximal operator
- why the improved performance Table 1 validates the decreased restrictiveness.

**Ethical Concerns:**

["NO or VERY MINOR ethics concerns only"]

**Final Justification:**

All my questions have been satisfactorily answered in the rebuttal, and I believe that this work merits acceptance to NeurIPS due to its strong mathematical foundation, high impact potential and excellent work quality.

**Limitations:**

Yes

**Quality:**

4

**Strengths And Weaknesses:**

Strengths:
- The idea of relaxing the current requirements on denoisers while still maintaining their proximal nature, making them interpretable as solutions to a minimization problem and providing strong convergence guarantees, is promising.
- The only essential change made to denoiser training is the addition of the two spectral norm terms, which can be used with any denoising architecture to encourage it to be CoCo.
- Using a CoCo denoiser is shown to be especially effective for Poisson inverse problems, where the likelihood is not guaranteed to be smooth globally. The competitive or superior results with this method, especially at with low signal, reflect the benefits of this property.

Weaknesses:
- For most of the theoretical development in the paper, it is assumed that $D_\sigma$ is a CoCo operator and guarantees for the convergence of CoCo-ADMM and COCO-PGD are derived. However, the training for $D_\sigma$ only *encourages* CoCo nature, doesn’t *enforce* it. It is possible to check the $\gamma$-cocoerciveness via the spectral norm $\|\|2\gamma J-I\|\|$ during training—however, it is not possible to check if the denoiser is “conservative enough”. The guarantees *require* conservativeness, however—so it is unclear, if even though the norm $\|\|J-J^T\|\|$ is close to zero, if the guarantee holds enough for it to be useful in reality. A natural next question to ask would be—how much could we relax the conservativeness to still ensure good enough convergence guarantees / proximal nature?
- In the evaluation on line 270, the paper claims that the PSNR values in Table 1 validate that “cocoercive and conservative properties are less restrictive for deep denoiser”. I’m not sure why this performance improvement can necessarily be thought of as an indication of lesser restriction—the performance improvement could also have been because of the different loss function.
- Appendix 13 claims that this is useful for real-time applications, even though 256x256 images take 5-10 seconds to process—I’m not sure if this claim makes sense.

Writing suggestions:
- Line 108: “epansive" $\rightarrow$ “expansive”
- Line 167: Why is there an expectation operator?
- Lines 204, 275, 283: Does the equation number need to be (18)?

---

> ### Author Rebuttal · Authors · 2025-07-26
>
> Dear reviewer aRzT,
>
> We sincerely appreciate your thoughtful review and constructive feedback. Your comments are highly valuable, and we have carefully considered each point in revising the paper. Our detailed responses are provided below.
>
> ### Responses to weaknesses:
>
> > 1.)  For most of the theoretical development in the paper, it is assumed that $D_\sigma$ is a CoCo operator and guarantees for the convergence of CoCo-ADMM and COCO-PGD are derived. However, the training for $D_\sigma$ only encourages CoCo nature, doesn’t enforce it. It is possible to check the $\gamma$-cocoerciveness via the spectral norm $\|2\gamma J-I\|_*$ during training—however, it is not possible to check if the denoiser is “conservative enough”. The guarantees require conservativeness, however—so it is unclear, if even though the norm $\|J-J^\mathrm{T}\|_*$ is close to zero, if the guarantee holds enough for it to be useful in reality. A natural next question to ask would be—how much could we relax the conservativeness to still ensure good enough convergence guarantees / proximal nature?
>
>
> We will use the word ``encourage`` rather than ``enforce``, because that spectral regularization can not guarantee the properties.
>
> In the experiments, we found that, if the denoiser is not conservative enough, the convergence can still be stable with fine-tuned parameters. This may be because that the Hamiltonian part of the denoiser is useless (and therefore sometimes harmless) for the denoising performance.
>
> To the best of our knowledge, there is no way to quantify how a non-conservative denoiser can still be ``good enough`` for being a proximal operator.
>
> We will include the discussions in the revised paper.
>
>
>
> > 2.)  In the evaluation on line 270, the paper claims that the PSNR values in Table 1 validate that “cocoercive and conservative properties are less restrictive for deep denoiser”. I’m not sure why this performance improvement can necessarily be thought of as an indication of lesser restriction—the performance improvement could also have been because of the different loss function.
>
> We agree with your comment. We will discuss carefully about the performance improvements. The performance improvement could also have been because of the different loss function.
>
>
> > 3.) Appendix 13 claims that this is useful for real-time applications, even though 256x256 images take 5-10 seconds to process—I’m not sure if this claim makes sense.
>
> We will remove the `real-time' expression in the revised paper.
>
>
>
> Please let us know if you have more suggestions. We are very willing to revise the paper based on your comments to further improve its quality.

---

> > ### Comment · Reviewer_aRzT · 2025-08-05
> >
> > Thank you for the response. All my questions have been satisfactorily answered, and I believe that this work merits acceptance to NeurIPS due to its strong mathematical foundation, high impact potential and excellent work quality.

---

> > > ### Author Response · Authors · 2025-08-05
> > > **Response to Reviewer aRzT**
> > >
> > > Dear Reviewer aRzT,
> > >
> > > Thank you for your positive feedback. You suggestions helped a lot to improve the paper. Please feel free to tell us if you have any other valuable advices.

---

### Official Review · Reviewer_G4Uk · 2025-06-28

**Clarity:** 2
**Significance:** 3
**Originality:** 3
**Rating:** 5
**Confidence:** 3

**Summary:**

This paper investigates plug-and-play (PnP) priors in the case of linear inverse problems with Poisson noise, which gives rise to a data fidelity term that is neither strongly convex nor L-smooth. It proposes to use co-coercive conservative (CoCo) denoisers in a PnP fashion, i.e., using a particular learning denoising network like the proximal operator of a classical optimization approach. CoCo denoisers are those that (implicitly) are the gradient of an underlying cost function and are $\gamma$-cocoercive (a property slightly stronger than being $1/\gamma$ Lipschitz). The authors prove convergence of the resulting PnP approach to a stationary point. As the constraints on the proposed denoiser are less restrictive, it yields a (slightly) better performance in the numerical results.

**Questions:**

- Please address the question of the setting (finite vs. infinite-dimensional) of (1) under weaknesses and possibly adapt the paper accordingly.
- Please answer the question why you chose the spectral norm to enforce symmetry of J.
- Do the values in table 1 (right) change if one optimizes for a point $x$ at which the norms are high?
- How large is the computational overhead in training the denoisers with spectral penalties involving its Jacobian?
- Have all (competing) methods been fine-tuned equally? Did you consider multiple runs with different random seeds?

**Ethical Concerns:**

["NO or VERY MINOR ethics concerns only"]

**Final Justification:**

The authors have addressed my concerns well and promised corresponding adaptations for the possible camera-ready version. I am increasing my rating.

**Limitations:**

I think two limitations could be stressed (as they do not appear in Section 6).
- All convergence guarantees are lost when only using a soft penalty on the denoiser.
- The fact that the approach (with strictly enforced constraints) is nice, but the interpretability gain of minimizing a cost function seems limited. The approach converges to a stationary point of a nonconvex cost function that involves a regularizes of which we only know the prox-operator (parameterized as a complex neural network). Advantages from classical (convex) approaches (such as error estimates) are lost.

**Paper Formatting Concerns:**

No concerns.

**Quality:**

3

**Strengths And Weaknesses:**

The paper is mostly well-written and well-motivated. The idea to impose the weakest possible constraints on the denoiser to maximize its performance while remaining in the regime of a provably convergent algorithm is very good, and the numerical results support the intuition that there can be gains in performance (although they remain very small).

Despite an extensive paper with a rich supplementary material and new theory, I also see some weaknesses.
(1) To my mind, the paper jumps a little between great mathematical generality (e.g., talking about Frechet differentials, suggesting that $V$ could be infinite dimensional) and implicitly making assumptions without introducing them, e.g. the Helmholtz decompositions needs a smoothness assumption, any time the transpose symbol is used, I assume we are in a finite dimensional setting or is this denoting the adjoint operator? Is the spectral norm the operator norm in the infinite-dimensional setting? Do all convergence results really hold in the infinite-dimensional setting as well? What assumptions do we need to make about $K$ in this case? If the convergence analysis holds in the finite-dimensional case only, I highly recommend writing the entire paper under this assumption and simplifying all parts that can be simplified. Moreover, the notion of the subdifferential and its Lipschitz continuity (Theorem 2.2.) should be introduced.
(2) The denoiser is conservative if the Jacobian is symmetric, i.e., $J=J^T$. The authors propose to regularize $||J-J^T||_*$, but it seems like any norm could be used. This is not discussed in the paper.
(3) In practice, the necessary assumptions are implemented via soft penalties only, such that all provable statements are lost. The demonstration that the assumptions are met is demonstrated "over 100 patches". Rather than being met on average, I recommend reporting the highest norm after actively optimizing for an $x$ at which a high norm is attained (similar to an adversarial example).
(4) The penalties require the computation and subsequent differentiation of the spectral norm of the Jacobian of the denoiser. Given that the denoiser is high-dimensional (mapping images to images), the Jacobian is extremely high-dimensional. While the appendix provides details about the computation, I think it is important to include a short statement on the computational expense of the training of the denoiser.
(5) Given the small differences in performance of the different methods, it is important to confirm that all training procedures are identical and to possibly report standard deviations.

---

> ### Author Rebuttal · Authors · 2025-07-26
>
> Dear reviewer G4Uk,
>
> We sincerely appreciate your thoughtful review and constructive feedback. Your comments are highly valuable, and we have carefully considered each point in revising the paper. Our detailed responses are provided below.
>
> ### Responses to weaknesses:
>
>
> Thank you for this valuable comments. Please note that, we fully agree with your suggestion that ``I highly recommend writing the entire paper under this assumption and simplifying all parts that can be simplified.'' We have revised the paper carefully, such that all settings are in a finite-dimensional space.
>
> But now, please allow us to respond to your questions. As answered below, most of the theoretical results in the paper still $\textbf{hold}$ in an infinite-dimensional setting.
>
>
>
> > 1.1)  To my mind, the paper jumps a little between great mathematical generality (e.g., talking about Frechet differentials, suggesting that $V$ could be infinite dimensional) and implicitly making assumptions without  introducing them, e.g. the Helmholtz decompositions needs a smoothness assumption, any time the transpose symbol is used, I assume we are in a finite dimensional setting or is this denoting the adjoint operator?
>
> Through out all theoretical parts in the paper, we were assuming a general real Hilbert space, which might be infinite-dimensional. We will include the smooth assumption for the Helmholtz decomposition. The transpose symbol means the adjoint operator in a Hilbert space $V$. When the space is finite dimensional like $\mathbb{R}^n$, the adjoint operator is the transpose.
>
> When comes to training, we will have to consider a finite dimensional space of course. So we agree that the paper should be written in a finite dimensional setting.
>
>
>
> > 1.2)  Is the spectral norm the operator norm in the infinite-dimensional setting?  If the convergence analysis holds in the finite-dimensional case only, I highly recommend writing the entire paper under this assumption and simplifying all parts that can be simplified.
>
> In an infinite-dimensional setting, $\|\cdot\|_*$ is the operator norm. When the space is finite-dimensional, it is just the spectral norm, that is, the largest singular value.
>
>
> > 1.3) Do all convergence results really hold in the infinite-dimensional setting as well?
>
> Thank you for this valuable comments. We have revised the paper, such that all results are included in a finite dimensional space.
>
>
> > 1.4) What assumptions do we need to make about $K$ in this case?
>
> When $V$ is just a general real Hilbert space, $K$ is a bounded linear operator, that is $K\in \mathscr{B}[V]$. When considering a finite dimensional space, this is just a matrix.
>
>
> > 1.4) Moreover, the notion of the subdifferential and its Lipschitz continuity (Theorem 2.2.) should be introduced.
>
> We will include the subdifferential and its Lipschitz continuity in the revised version. Thank you so much for such a detailed comment.
>
>
>
>
> > 2.) The denoiser is conservative if the Jacobian is symmetric, i.e., $J=J^\mathrm{T}$. The authors propose to regularize $\|J-J^\mathrm{T}\|_*$, but it seems like any norm could be used. This is not discussed in the paper.
>
> Special thanks for this valuable feedback! We choose to use the spectral norm, mainly because that it is has low computational cost. Due to the power iterative method, one can calculate the spectral norm very easily, without actually calculate the Jacobian, see Appendix A.5 for details. For other norms, we do not know whether there exist similiar algorithms to calculate them. We will include this discussion in the revised paper.
>
>
>
> > 3.)  In practice, the necessary assumptions are implemented via soft penalties only, such that all provable statements are lost. The demonstration that the assumptions are met is demonstrated "over 100 patches". Rather than being met on average, I recommend reporting the highest norm after actively optimizing for an $x$ at which a high norm is attained (similar to an adversarial example).
>
> Now the highest norm for $\|J-J^\mathrm{T}\|_*$ is reported below. Please note that the highest norm for cocoercivity has already been reported in Table 1, and thus omitted here.
>
>
> $ $|  15  | 25 |  40 | Norms
> :----: | :----: | :----: | :----: | :----:
> DRUNet | 3.697e+2  |   8.107e+4  | 2.077e+4  | $\|J-J^\mathrm{T}\|_*$
> 0.5-CoCo-DRUNet | 1.841e-5  |  3.135e-6   | 1.978e-5  | $\|J-J^\mathrm{T}\|_*$
> 0.5-CoCo-DRUNet |  6.629e-5 |   2.754e-6  |  7.827e-4 | $\|J-J^\mathrm{T}\|_*$
>
> This table will be included in the revised paper upon acceptance.
>
> > 4.)  The penalties require the computation and subsequent differentiation of the spectral norm of the Jacobian of the denoiser. Given that the denoiser is high-dimensional (mapping images to images), the Jacobian is extremely high-dimensional. While the appendix provides details about the computation, I think it is important to include a short statement on the computational expense of the training of the denoiser.
>
> Thank you for this comment. Since the rebuttal window does not allow extra PDF or images, we report by words the computational cost. We first pre-train a DRUNet without spectral regularization. After that, we post-train the DRUNet with CoCo regularization. This is for saving training time.
>
> Compared with the training for DRUNet without any spectral regularization, for each epoch, the computational time for training CoCo is about $12$ times larger than training DRUNet without spectral regularizations. For the GPU memory cost, DRUNet is about $2080$ MB, and CoCo is about $9360$ MB.
>
>
>
> > 5.) Given the small differences in performance of the different methods, it is important to confirm that all training procedures are identical and to possibly report standard deviations.
>
> We will state in the revised paper, that all training procedures for all denoisers are identical. In the revised paper, we will also report standard deviations with different random seed for denoising performance comparison.
>
>
> ### Responses to questions:
>
> > Please address the question of the setting (finite vs. infinite-dimensional) of (1) under weaknesses and possibly adapt the paper accordingly.
>
> Please refer to the responses to the weaknesses 1.1.
>
> > Please answer the question why you chose the spectral norm to enforce symmetry of $J$.
>
> Please refer to the responses to the weaknesses 2.
>
> > Do the values in table 1 (right) change if one optimizes for a point $x$ at which the norms are high?
>
> Unfortunately, yes. Thus we agree with your suggestions, and report the maximum norms, see the responses to the weaknesses 3.
>
> > How large is the computational overhead in training the denoisers with spectral penalties involving its Jacobian?
>
> Unfortunately, pretty large: the computational cost for CoCo is 12 times larger than the training for DRUNet, see the responses to the weaknesses 4. As a result, we pre-trained a DRUNet first, and then post-train it with CoCo regularizations to save the training time.
>
>
> > Have all (competing) methods been fine-tuned equally? Did you consider multiple runs with different random seeds?
>
> We have tried our best to fine tune each method, such that they reach the best PSNR values. For the random seed, for a fair comparison, we set the same random seed for all methods.
>
> ### Responses to limitations:
>
> Thanks for your suggestions on the limitations. We will address them in Section 6 in the revised paper.
>
>
>
> Please let us know if you have more suggestions to polish the paper and raise the rating. We are very willing to revise the paper based on your comments to further improve its quality.

---

> ### Comment · Reviewer_G4Uk · 2025-08-05
>
> Thanks for the detailed response! I will increase my rating to an accept.
>
> Just to make sure my comment regarding the highest |J-J^T| was clear: Since the Jacobian depends on the point where you evaluate it, I was thinking about approximately solving \max_x |J(x)-J^\mathrm{T}(x)| (not just sampling x from a batch). Is this what you did for the new table? If so, how did you approximate the solution of the maximization problem?

---

> > ### Author Response · Authors · 2025-08-05
> > **Response to Reviewer G4Uk**
> >
> > Dear Reviewer G4Uk,
> >
> > Thank you for your comments, and raising your rating!
> >
> > We were indeed reporting the highest  $\|J-J^\mathrm{T}\|_*$, because the value will change if $x$ changes. However, we have to report the max value on a batch. Because there are infinite $x$ even in a finite dimensional space $V$. So we just choose a batch, and record all the norms, and report the maximum in the new table.
> >
> > In the training process, we did not solve this maximization problem (that is, we did not solve an $x$ that maximizes the norm), but chose to penalize $\|J(x)-J^\mathrm{T}(x)\|_*$ at any current training image $x$. Frankly speaking, I think penalizing the maximum norm is more reasonable. But now I have no clue how to do it. We will state it clearer in the main text if the paper gets accepted.
> >
> > Please tell us if you have more questions. We are more than willing to revise the paper according to your advice.

---

### Official Review · Reviewer_UHTR · 2025-06-30

**Clarity:** 2
**Significance:** 3
**Originality:** 4
**Rating:** 4
**Confidence:** 2

**Summary:**

This paper proposes a new class of deep denoisers, called CoCo denoisers, for solving Poisson inverse problems via plug-and-play (PnP) methods. Unlike prior works that require restrictive non-expansiveness, the proposed CoCo denoisers are both cocoercive and conservative, making them more flexible while still ensuring theoretical convergence. The authors introduce a novel training scheme using spectral and Hamiltonian regularization to enforce these properties. Theoretical analysis shows CoCo denoisers are proximal operators of weakly convex functions, and experiments demonstrate strong performance on Poisson image restoration tasks.

**Questions:**

My concerns are outlined in the Weaknesses section. To be responsible, I must admit that I am not fully familiar with some of the theoretical components presented in this paper, and thus I am not in a position to make a definitive judgment. To be cautious, I am currently assigning a borderline rating. However, if the authors can address some of my questions during the rebuttal phase, I would be open to raising my rating.

**Ethical Concerns:**

["NO or VERY MINOR ethics concerns only"]

**Final Justification:**

First of all, we must reiterate that we are not very familiar with some of the theoretical aspects presented in this paper. We can only evaluate the work based on the parts we understand. We believe the paper holds certain theoretical value. However, its weaknesses are also clear: the theoretical presentation is almost entirely based on formulas and theorems, lacking visual or intuitive illustrations, which makes it somewhat difficult to follow for readers who are not well-versed in this field. That said, we acknowledge that conducting in-depth theoretical research is a challenging task and should be encouraged. Therefore, we have decided to raise our rating to "borderline accept," while keeping our confidence level unchanged. We kindly ask the area chair to consider our decision as a partial reference only.

**Limitations:**

yes

**Quality:**

4

**Strengths And Weaknesses:**

Strengths

(1) The idea of combining cocoerciveness and conservativeness in denoiser design is quite novel. Using Helmholtz decomposition to interpret the denoising behavior adds a fresh perspective that's both theoretically grounded and intuitive.

(2) The authors provide solid theoretical results, proving that their denoiser acts as a proximal operator of a weakly convex function. This gives convergence guarantees even under the nonconvex and nonsmooth setting of Poisson inverse problems.

(3) The experimental results are convincing. On several Poisson imaging tasks, the proposed methods outperform strong baselines like DPIR and Prox-DRS in both PSNR and SSIM.

Weaknesses

(1) While the paper proposes cocoerciveness as a relaxed alternative to non-expansiveness, it is unclear why this particular relaxation is theoretically sufficient and practically better. What exactly fails when the operator is not cocoercive, and how tight is this condition?

(2) The theoretical guarantee that the CoCo denoiser is a proximal operator relies on both cocoerciveness and exact conservativeness (i.e., symmetry of the Jacobian). In practice, both are only enforced softly during training. Does this soft enforcement suffice for convergence? What happens when these conditions are only approximately satisfied?

(3) The use of Helmholtz decomposition to interpret denoiser geometry is elegant, but relies on assumptions that may not hold in practice. Is the decomposition stable across different architectures and noise levels? How reliable is the connection between the anti-symmetric part and “useless” Hamiltonian directions in high-dimensional deep networks?

(4) The convergence analysis (e.g., in Theorem 4.1) introduces a parameter $t<t_0(\gamma)$ where $t_0$ is the root of a cubic equation. However, the paper does not give much intuition on what range of $t$ works best in practice, or how sensitive the algorithm is to the choice of $\gamma$ and $t$. A more detailed discussion or ablation would help.

(5) Although the paper presents a rich and rigorous theoretical framework, the presentation is overwhelmingly formal and math-heavy, with long sections dominated by formulas and lemmas. It may be very difficult for readers—especially those not from optimization theory—to grasp the main ideas. Some intuitive explanation, diagrams, or simplified takeaways would be highly beneficial.

---

> ### Author Rebuttal · Authors · 2025-07-26
>
> Dear reviewer UHTR,
>
> Thank you so much for your valuable feedback and insightful comments. We sincerely appreciate your encouraging words and constructive critique; both are extremely valuable. We have given careful consideration to your points and are focused on refining the paper. Our detailed responses are provided below as part of our pursuit of excellence.
>
> ### Responses to weaknesses:
> > 1.) While the paper proposes cocoerciveness as a relaxed alternative to non-expansiveness, it is unclear why this particular relaxation is theoretically sufficient and practically better. What exactly fails when the operator is not cocoercive, and how tight is this condition?
>
> Special thanks for this comment! This question is very valuable, and we will discuss this not only here, but also in the revised paper.
>
> There basically two ways to use a prior in the context of PnP. If we use it in a forward way, for example to use it in RED, to the best of our knowledge, the weakest assumption is pseudo-contractiveness as in [71]. But if we use it in a backward way, for example this paper uses it as a prox, the weakest assumption (in terms of a weakly convex prior) has been proven to be `Lipschitz + be the gradient of some convex potential' in [22]. [22] established the equivalence between 'the denoiser is a proximal operator of some weakly convex prior', and 'the denoiser is Lipschitz and is the gradient of some convex potential'. Please note that, 'the denoiser is Lipschitz and is the gradient of some convex potential' can be equivalently rewritten as 'the denoiser is conservative and cocoercive': cocoerciveness ensures that the denoiser is Lipschitz and monotone; combined with conservativeness, the denoiser is then a gradient of a convex potential. The gradient of a convex potential is a special case of monotone operators.
>
> The proof for this equivalence was not included in the paper. Currently, in Theorem A.5, we have proved that if the denoiser is conservative and cocoercive, it is a prox of some weakly convex prior. We will include the equivalence in the revised version.
>
> > What exactly fails when the operator is not cocoercive, and how tight is this condition?
>
> As stated above, if we want the denoiser to be a prox of some weakly convex prior, the cocoercive and conservativeness are required. The condition is tight, and actually sufficient and necessary.
>
>
> > 2.) The theoretical guarantee that the CoCo denoiser is a proximal operator relies on both cocoerciveness and exact conservativeness (i.e., symmetry of the Jacobian). In practice, both are only enforced softly during training. Does this soft enforcement suffice for convergence? What happens when these conditions are only approximately satisfied?
>
> We softly enforce cocoerciveness and conservativeness by spectral regularization technique. The established convergence requires the two properties. In experiments, we found that, if the denoiser is not conservative enough, the convergence can still be stable with fine-tuned parameters. This may be because that the Hamiltonian part of the denoiser is useless (and therefore sometimes harmless) for the denoising performance.  We will include this in the revised paper.
>
> > 3.) The use of Helmholtz decomposition to interpret denoiser geometry is elegant, but relies on assumptions that may not hold in practice. Is the decomposition stable across different architectures and noise levels? How reliable is the connection between the anti-symmetric part and “useless” Hamiltonian directions in high-dimensional deep networks?
>
> The decomposition is stable with different architechtures and noise levels, as long as the network output is differentiable with respect to the input, such that $J$ is computable. When the dimension is high, the connection between the anti-symmetric part and “useless” Hamiltonian directions is actually illustrated briefly in the equation blow Line 172. This means that the inner product of $D(y)-x$ and $\nabla_y \mathcal{H}(y)$ is zero, and thus the two vector is orthogonal. This further means that, if we consider the continuous dynamics, $\dot{y}=-\nabla_y \mathcal{H}(y)$, $\mathcal{H}(y(t))\equiv Constant$. That is, the dynamics preserve the Hamiltonian functional, if the denoiser only has Hamiltonian part. Since the Hamiltonian functional is a typical denoising loss, this means that a Hamiltonian field does not contribute to the denoising.
>
> We will include more discussion on high dimensional cases in the revised paper.
>
> > 4.) The convergence analysis (e.g., in Theorem 4.1) introduces a parameter $t<t_0(\gamma)$, where $t_0$ is the root of a cubic equation. However, the paper does not give much intuition on what range of $t$ works best in practice, or how sensitive the algorithm is to the choice of $\gamma$ and $t$. A more detailed discussion or ablation would help.
>
> Since the rebuttal window does not allow PDF or inserting images, we only report the PSNR here. We have conducted oblation study with different $\gamma$ and $t$. We consider deblurring for the image `0005.png' from CBSD68 and kernel 8 from Levin's dataset. We set the Poisson noise level to be $50$. We run the algorithm for 500 iterations to ensure the convergence.
>
> When $\gamma=0.25$, $t_0(\gamma)=\frac{1}{3}$. Thus $t<\frac{1}{3}$.
>
>
> $ $|  $t$   | $0.3333$ |  $0.3$ | $0.2$ | $0.1$ | $0.001$
> :----: | :----: | :----: | :----:| :----: | :----: | :----:
> CoCo-ADMM |PSNR    | 25.33    | 25.31 |   25.30 | 25.24 | 25.16
> CoCo-PEGD |PSNR    | 25.32    | 25.30 |  25.27 | 25.22 | 24.98
>
> When $\gamma=0.5$, $t_0(\gamma)\approx 0.3761$. Thus $t\le 0.3761$.
>
> $ $|  $t$   | $0.3761$ |  $0.3$ | $0.2$ | $0.1$ | $0.001$
> :----: | :----: | :----: | :----: | :----: | :----: | :----:
> CoCo-ADMM |PSNR    | 24.88    |  24.87 | 24.79 | 24.64 | 24.47
> CoCo-PEGD |PSNR    | 24.84    |   24.82 | 24.71 |24.54 | 24.35
>
> From the two tables, it is clear that when $t$ is not too small, the PnP performance is not sensitive to the choice of $t$. However, different $\gamma$ has significant influence on the PSNR performance. This is because that a larger $\gamma$ means a more restrictive assumption on the denoiser, and therefore results in a worse performance. In practice, we recommend to set $t\approx t_0(\gamma)$, with $\gamma=0.25$. No significant denoising improvement is observed when $\gamma<0.25$.
>
>
>
>
> > 5.) Although the paper presents a rich and rigorous theoretical framework, the presentation is overwhelmingly formal and math-heavy, with long sections dominated by formulas and lemmas. It may be very difficult for readers—especially those not from optimization theory—to grasp the main ideas. Some intuitive explanation, diagrams, or simplified takeaways would be highly beneficial.
>
> Thank you for this valuable feedback. After thorough reading our own paper, we have to admit that, the paper is indeed a little math-heavy. We will revise the paper carefully, and add some intuitive explanation, diagrams, simplified takeaways, and proof sketches to make the paper easier to read.
>
>
>
> Please let us know if you have more suggestions to polish the paper and raise the rating. We are very willing to revise the paper based on your comments to further improve its quality.

---

> > ### Comment · Reviewer_UHTR · 2025-08-02
> >
> > Thank you for your thoughtful response. Your reply has addressed most of our concerns, and we will consider increasing our score. However, we still have a few suggestions: although the paper is theoretically strong, we recommend providing some diagrams to intuitively illustrate your theory, rather than relying solely on equations. After all, if your paper is accepted, it will likely be read by many beginners or researchers who are not yet familiar with this field.

---

> > > ### Author Response · Authors · 2025-08-03
> > > **Reply to Reviewer UHTR**
> > >
> > > Dear Reviewer UHTR,
> > >
> > > Thank you so much for your feedback! We agree with your suggestions. We will include some diagrams to illustrate intuitively not only the characterization theorem, but also the convergence  conditions. We will try to make the paper easier to read, especially for the readers not familiar with this field.
> > >
> > > Thank you again. Please feel free to tell us if you have more helpful suggestions.

---

### Official Review · Reviewer_8mzP · 2025-07-03

**Clarity:** 3
**Significance:** 2
**Originality:** 1
**Rating:** 4
**Confidence:** 3

**Summary:**

This work is part of the broader topic of constraining denoisers used in plug-and-play algorithms for solving inverse problems in order to recover convergence guarantee.
Previous convergence guarantees for PnP methods often come with constraints on the non-expansiveness and the conservativeness of the learned denoiser, with a price on the empirical performance compared to unconstrained denoisers.
Besides, most existing work assume gradient Lipschitzness of the datafit which is not applicable regarding Poisson noisy degradations with non smooth KL datafit.
The goal of this paper is to replace the non-expansiveness enforcement by a weaker cocoercivity constraint while having a setup that can effectively handle Poisson inverse problems.

**Questions:**

1. A first thing that particularly struggles me is the absence of reference to the paper  Convergent Bregman Plug-and-Play Image Restoration for Poisson Inverse Problems (Hurualt and al.). While the methods from this paper do are cited, the reference is lacking. Besides, I think the positionning of this paper should be clarified: as far as I understand, Hurault's paper also use a conservative denoiser (it is modeled as a gradient by design) and show that their denoiser is a (Bregman) prox. The main difference is that what is done in this paper is that it uses Gaussian denoisers and classical Euclidian prox (compared to Bregman prox): can the authors provide more comments on their method compared to the choice done in Hurault's work which uses a more well suited Bregman geometry.

2. I may have missed something but is the Helmholtz decomposition really needed ? I feel like the result needed for convergence guarantees is just Proposition 2. of [22] which basically states that being the prox of a weakly convex function is equivalent to be Lipschitz + be some gradient of a convex potential. I don't see how Helmholtz decomposition relates with being the gradient of a *convex* potential.

3. As the authors leverage prox characterization of [22], the authors should be able to compute the implicit regularization which would give the authors a way to plot the objective function decrease (equation (15)) during the iterations of the algorithm.

4. On the algorithms:
-  CoCo-PEGD: the authors actually smooth the datafidelity term (ensuring it has Lipschitz gradient): this seems quite contractory regarding the first motivation of the work which is to handle non-smooth datafits.

**Ethical Concerns:**

["NO or VERY MINOR ethics concerns only"]

**Final Justification:**

I have raised my score to weak accept following the rebuttal. The authors provided convincing explanations for previously unclear points. I believe the main strength of the paper lies in its specialization to Poisson inverse problems, which is a relevant contribution. I do not provide a higher score as I believe the soft regularisations used to enforce co-coercivity and conservativity of the denoiser are not particulary novel and in practice do not provide any guarantees (these constraints are softly enforced on training points and not on the whole space). Moreover, the main novelty of the approach essentially lies in relaxing the denoiser to be the prox of a weakly convex penalty instead of being the prox of a convex function: while this is an interesting direction, it still builds on several known ideas and similar formulations from prior work.

**Limitations:**

Yes

**Quality:**

2

**Strengths And Weaknesses:**

**Strengths**

The overall topic of recovering classical optimization convergence guarantees by enforcing some well-chosen properties on the learned denoiser is interesting. Among them, cocoercivity is a new and appropriate choice. The emprical results do improve on the other mtehods for Poisson restoration.


**Weaknessess**

While the general idea of the paper is novel and interesting; I feel that at the end it reduces to some soft enforcement of the constraints on the denoisers which are already seen in the literature:
- cocoercivity comes down to relaxing by a parameter $\gamma$ the firmly non-expansiveness soft contraint of Terris and al [66]
- enforcing conversativeness of the denoiser (i.e. having a denoiser which is a gradient) can also be found in the literature (RED, Gradient Step denoisers)
Besides, the proof techniques for convergence strongly rely on the prox characterization of [22] which has already been used a lot in Hurault's works.

---

> ### Author Rebuttal · Authors · 2025-07-26
>
> Dear reviewer 8mzP,
>
> Thank you so much for your professional feedback and suggestions on the originality and theory. We sincerely appreciate your positive and negative comments, which are extremely helpful. We have carefully considered your comments and are committed to refining our paper. Below, you will find our detailed response as we strive for excellence.
>
> ### Responses to weaknesses:
> > 1.) Cocoercivity comes down to relaxing by a parameter $\gamma$ the firmly non-expansiveness soft constraint of Terris and al. [66].
>
> Cocoercivity is a much broader (and weaker) concept than firm non-expansiveness, and residual non-expansive. In particular, if $\gamma=1$, $\gamma$-cocoercivity is equivalent to firm non-expansiveness; if $\gamma=0.5$, $\gamma$-cocoercivity is equivalent to residual non-expansiveness. Terris et al. proposed to softly enforce the firm non-expansiveness of the denoiser. By doing so, the denoiser is proved to be a resolvent of some monotone operator prior. If further assumed conservativity, this means that the learned prior is convex. However, as reported by many past papers [28,71], enforcing such constraints significantly compromises the denoising performance and PnP restoration. This paper relaxes this contraint to cocoercivity. If the denoiser is $\gamma$-cocoercive with $\gamma\in(0,1)$, the denoiser is actually a resolvent of some weakly monotone operator prior. If further assumed conservativity, this means that the learned prior is weakly convex. Overall, this paper adopted the soft constraints technique by Terris et al., and extended it to a weaker concept.
>
> Please note that, compared to the pioneer work by Terris and Pesquet et al., this paper has the following significant improvements:
>
> (1) The learned prior is weakly convex, enabling much better PnP restoration performance. In Table 2, compared to RMMO-DRS, the proposed methods has about 1dB increase in PSNR.
>
> (2) We proved the convergence of PnP-ADMM with an implicit weakly convex prior, and a non-strongly convex, non-smooth data fidelity prior.
>
> > 2.) Enforcing conservativeness of the denoiser (i.e. having a denoiser which is a gradient) can also be found in the literature (RED, Gradient Step denoisers). Besides, the proof technique for convergence strongly rely on the prox characterization of [22] which has already been used a lot in Hurault's works.
>
> Gradient Step denoiser (GS-DRUNet) is indeed a great way to enforce the conservativeness. However, please note that, the training and the inference procedure of GS-DRUNet both need to calculate Jacobian-vector product, because there is always a Jacobian term in GS-DRUNet. This increases the training cost.
>
> This paper wants to propose an alternative way to learn a conservative denoiser. We introduced the Helmholtz decomposition to study the denoising geometry, and showed that the Hamiltonian part is useless for denoising, and thus an ideal denoiser should be conservative. Besides, this decomposition also inspires us how to train a conservative denoiser without modifying the network architechture as in GS-DRUNet.
>
> For the proof, we do utilize the great work [22] by Gribonval and Nikolova. Although Hurault has also use it to characterize the denoiser, there is an important difference between our paper and Hurault's paper. The Lipschitz assumption on the denoiser in Hurault's paper is residual non-expansiveness. This assumption is still restrictive, especiall when the noise level is large. In our paper, the assumption is relaxed to be cocoercive with even small $\gamma$. This relaxation $\textbf{fully}$ utilized the result in [22]. Mathematically speaking, since the `Lipschitz + be some gradient of a convex potential' can be equivalently rewritten as 'the denoiser is conservative and cocoercive', the proposed CoCo denoiser is a $\textbf{tight}$ utilization of the characterization in [22], and therefore provides better PnP performance.
>
>
>
>
> ### Responses to questions:
>
> 1. We apologize for this careless mistake: in fact, in line 85-88 in the introduction, we have already talked about Bregman-PnP by Hurault et al.. However, we cited the wrong paper. We actually wanted to cite the paper `Convergent Bregman Plug-and-Play Image Restoration for Poisson Inverse Problems' published in Neurips. Besides, in the experiments, we have already compared the proposed method with Bregman-PnP (termed as B-PnP in the paper), see Table 2 and Figure 2.
>
> Besides, thank you for your clarification! We totally agree with your statement that, we proposed to use a Gaussian denoiser as a prox rather than a Bregman prox. Compared to Bregman-PnP, the proposed CoCo-PnP has the following advantages:
>
> (1) In the context of PnP, people use Gaussian denoisers mainly because that Gaussian denoisers are easier to get than other type denoisers. However, as stated in the introduction, there is a lacking literature on using Gaussian denoisers as prox to solve Poisson inverse problems. Our paper aims to solve this problem with Gaussian denoiser as a proximal operator of some implicit weakly convex prior.
>
> (2) As reported in Pages 10 and 30 in the Appendix of the paper `Convergent Bregman Plug-and-Play Image Restoration for Poisson Inverse Problems', for large noise levels, Bregman denoiser does not perform well. Besides, when applied to deblurring with Poisson noises, Bregman-PnP does not outperform existing methods. We have also observed this in the experiments, see Table 2 and Figure 2 for example. Compared to Bregman-PnP (termed B-PnP in our paper), the proposed B-PnP performs significantly better. This suggests that using Gaussian denoisers in Poisson inverse problems is more suitable.
>
> 2. The Helmholtz decomposition is useful to not only interpret the denoising geometry, but also inspire us the design the loss functions, such that the learned denoiser is encouraged to be conservative. Proposition 2 does only require the denoiser to be Lipschitz and be some gradient of a convex potential. But, please note that, these two requirements are equivalent to that `the denoiser is conservative and cocoercive'. For the proof, please see line 596-600 in our paper, in which we have proved that the denoiser is a gradient of some convex potential.
>
> Intuitively speaking, the `convexity' of the potential mainly comes from the cocoercivity of the denoiser. Please note that cocoercivity actually requires the denoiser to be a monotone operator. The gradient of a convex potential is a special case of monotone operators.
>
> 3. Unfortunately, we are not able to compute it. This is because that, we can not access the convex potential of the denoiser. Different from the great pioneering works by Hurault et al., the proposed denoiser has an implicit potential. According to Theorem 3 in [22], or Proposition 3.1 by Hurault et al. in [29], in order to compute the prior, we need to access the potential. Therefore, we chose to report the PSNR and relative error curves in Figures 2-3, 6-7. These figures validates the convergence.
>
> 4. For a differentiable data fidelity term, in the context of operator splitting algorithms, there are basically two ways to use it. We can use the fidelity in a backward way, that is, to use its proximal operator like in PnP-ADMM. Since the fidelity is convex, the proximal operator of such fidelity has good property: it is firmly non-expansive. We can also use the fidelity in a forward way, that is, to use its gradient like in PGD or FBS. But since the fidelity is not smooth, we have to smooth it by considering its Moreau envelope. We chose to include this algorithm not only for completedness of the two ways to use the fidelity, but also that because we wanted to show that, sometimes, the first motivation can not be solved by simply smoothing the non-smooth fidelity. In fact, as reported in Table 2, CoCo-ADMM performs better than CoCo-PEGD: this is because that the fidelity is changed to its envelope, and there can not fully reflect the Poisson distribution.
>
>
>
> Thank you again for such professional comments on our paper. Please let us know if you have more suggestions. We are very willing to revise our paper based on your comments.

---

> > ### Author Response · Authors · 2025-08-03
> > **Waiting for Reply from Reviewer 8mzP**
> >
> > Dear Reviewer 8mzP,
> >
> > Thank you very much for your valuable comments. We have carefully addressed each of your suggestions point by point. As we have not received any further response, we would like to kindly ask if you have any additional questions or concerns regarding the paper. We are more than willing to revise the manuscript further based on your guidance.

---

> > ### Comment · Reviewer_8mzP · 2025-08-03
> > **Response to authors**
> >
> > Thank you for your detailed answer.
> >
> > Regard response to weaknesses:
> > - I appreciate your clarification, notably the thight use of the prox characterisation, which I think deserve to be stated in the main paper.
> > - You say that the Gradient Step denoiser has a Jacobian vector product term that increases the cost, but the soft regularizations on the Jacobian for the CoCo denoiser also likely increase the training cost. It would be very helpful to include a comparison of time and memory usage for each method in the pape
> >
> > Regarding response to question (1.) I did not found B-PnP  in Table 2 and Figure 2, could you please clarify whether B-PnP was included in these results, and if not, could you provide the results here?
> >
> > Overall, I think the methodology is rigorous with an interesting specification for Poisson problems. Lots of numerical expriments and an extensive benchmark are provided which is a nice contribution. With the clarifications above addressed, I would be inclined to increase my score.

---

> > > ### Author Response · Authors · 2025-08-04
> > > **Response to Reviewer 8mzP**
> > >
> > > Dear Reviewer 8mzP,
> > >
> > > Thank you for your valuable feedback.
> > >
> > > $\bullet$ We will state the tight use of the prox characterization in the main text.
> > >
> > > $\bullet$ In the training process, CoCo does need to calculate the Jacobian-vector product. However, after the training process, CoCo serves just like a regular denoiser, and does not need to calculate the Jacobian-vector product. Note that GS-DRUNet still need to calculate the Jacobian-vector product even after the training process. This therefore results in extra computational cost. We have already included the computational time in Table 6 in Page 28. Below we also report the GPU memory cost by each methods during PnP restoration.
> > >
> > > $ $|  B-RED   | Prox-DRS |  SNORE | DiffPIR | DPS | CoCo-ADMM | CoCo-PEGD
> > > :----: | :----: | :----: | :----:| :----: | :----: | :----: | :----:
> > > Iteration  | 300   | 100    | 200 | 100   | 1000 | 50 |    100
> > > Time (s) | 42.5   |   13.6  | 74.1  | 11.8  | 177 | 5.24 | 9.08
> > > Memory (MB) |  4946  | 3982   |  3624  | 2998  | 4694 | 2304 | 2304
> > >
> > > It can be seen in this table that, compared to other closely-related methods, the proposed CoCo-PnP methods need to least time for convergence, and least memory for inference.
> > >
> > > For question 1, we refered B-PnP method as B-RED as in Table 2 and Figure 2. B-RED denotes Bregman Regularization-by-Denoising, as named in Section 4.2 in the original paper of Hurault's et al..
> > >
> > > Please tell us if you have more valuable suggestions to improve the paper. We are very willing to revise it according to your advices.

---

> > > ### Author Response · Authors · 2025-08-05
> > > **Waiting for further comments from Reviewer 8mzP**
> > >
> > > Dear Reviewer 8mzP,
> > >
> > > Thank you very much for your valuable comments. We have carefully addressed each of your new concerns point by point. As the discussion period is about to end, we would like to kindly ask if you have any further questions or suggestions regarding the paper. We would be more than happy to revise the manuscript based on your advice.

---

### Comment · Area_Chair_kuXa · 2025-08-01
**Reminder: Please Engage in Author Response and Discussion**

Dear Reviewers,

Thanks again for your time and effort as NeurIPS reviewers.

As we enter the author response and discussion phase, I kindly ask that you take the time to **carefully read the author responses, especially any replies directed specifically to your comments**. Please try to respond to the authors **as early as possible during this discussion window**, so that we have enough time for a meaningful back-and-forth discussion.

In particular, I encourage you to:

1. **Acknowledge that you have read the author response, and engage with any new arguments, clarifications, or evidence provided**;

2. **Discuss any points of disagreement openly and constructively, with both the authors and your fellow reviewers**;

3. **Read the other reviews and responses, and participate in the broader discussion when you notice divergence of opinion or incomplete coverage of important aspects.**

Your thoughtful engagement during this phase is critical to ensuring a fair and informed decision process. Thank you very much!

Best regards,

AC

---

### Decision · Program_Chairs · 2025-09-17

**Decision:**

Accept (poster)

**Comment:**

**Meta-Review:**

This paper presents a novel approach to Poisson inverse problems using cocoercive conservative (CoCo) denoisers in plug-and-play (PnP) methods. The authors offer rigorous theoretical guarantees and demonstrate strong empirical performance, particularly in Poisson image restoration tasks.

### **Strengths:**

* **Novelty and Theoretical Contribution:** The use of cocoercivity as a relaxation of non-expansiveness for denoisers in PnP methods is a significant theoretical advancement. The convergence proof for PnP with weakly convex priors is well-founded.
* **Empirical Validation:** The method shows strong performance in comparison to closely related methods, particularly in challenging scenarios like Poisson inverse problems.
* **Well-Structured Paper:** The methodology is clearly presented with extensive experimental results supporting the theoretical claims.

### **Weaknesses:**

* **Soft Enforcement of Conditions:** The soft enforcement of cocoercivity and conservativeness raises concerns about the **practical sufficiency** of the convergence guarantees. The authors addressed this in the rebuttal, but further clarification is needed on how this affects real-world performance.
* **Helmholtz Decomposition:** While insightful, its **applicability across architectures and noise levels** in high-dimensional settings could be further explored.
* **Interpretation of Performance Gains:** Performance improvements could also be attributed to factors like **loss function changes**, which the authors should clarify further.
* **Math-Heavy Presentation:** The paper could benefit from more **intuitive explanations** and **diagrams** to make the theoretical framework more accessible.

### **Discussion & Rebuttal:**

The authors responded well to reviewer concerns, providing **clarifications** on the theoretical guarantees, the Helmholtz decomposition, and experimental design. However, **further refinement in presentation** (especially for non-experts) and additional validation are still needed.

### **Final Recommendation:**

**Accept**, but with **strong recommendations for revisions**. The theoretical contributions and empirical results are strong, but addressing the practical implications of the **soft enforcement** of conditions and providing more accessible explanations and comparisons will enhance the paper’s impact.